# INNOVATORBENCH: EVALUATING AGENTS' ABILITY TO CONDUCT INNOVATIVE LLM RESEARCH

**Yunze Wu**[1,3]*, **Dayuan Fu**[2,3]*, **Weiye Si**[1,3], **Zhen Huang**[2,3], **Mohan Jiang**[1,2], **Keyu Li**[1,2],
**Shijie Xia**[1,2,3], **Jie Sun**[2,3], **Tianze Xu**[1,3], **Yang Xiao**[3], **Pengrui Lu**[1,2,3], **Xiaojie Cai**[1,3],
**Lyumanshan Ye**[1,3], **Wenhong Zhu**[1,2], **Xiangkun Hu**[2,3], **Pengfei Liu**[1,2,3]†
[1]Shanghai Jiao Tong University, [2]Shanghai Innovation Institute, [3]GAIR

## ABSTRACT

AI agents could accelerate scientific discovery by automating hypothesis formation, experiment design, coding, execution, and analysis, yet existing benchmarks probe narrow skills in simplified settings. To address this gap, we introduce InnovatorBench, a benchmark-platform pair for realistic, end-to-end assessment of agents performing Large Language Model (LLM) research. It comprises 20 tasks spanning Data Construction, Filtering, Augmentation, Loss Design, Reward Design, and Scaffold Construction, which require runnable artifacts and assessment of correctness, performance, output quality, and uncertainty. To support agent operation, we develop ResearchGym, a research environment offering rich action spaces, distributed and long-horizon execution, asynchronous monitoring, and snapshot saving. We also implement a lightweight ReAct agent that couples explicit reasoning with executable planning using frontier models such as Claude-4, GPT-5, GLM-4.5, and Kimi-K2. Our experiments demonstrate that while frontier models show promise in code-driven research tasks, they struggle with fragile algorithm-related tasks and long-horizon decision making, such as impatience, poor resource management, and overreliance on template-based reasoning. Furthermore, agents require over 11 hours to achieve their best performance on InnovatorBench, underscoring the benchmark's difficulty and showing the potential of InnovatorBench to be the next generation of code-based research benchmark.

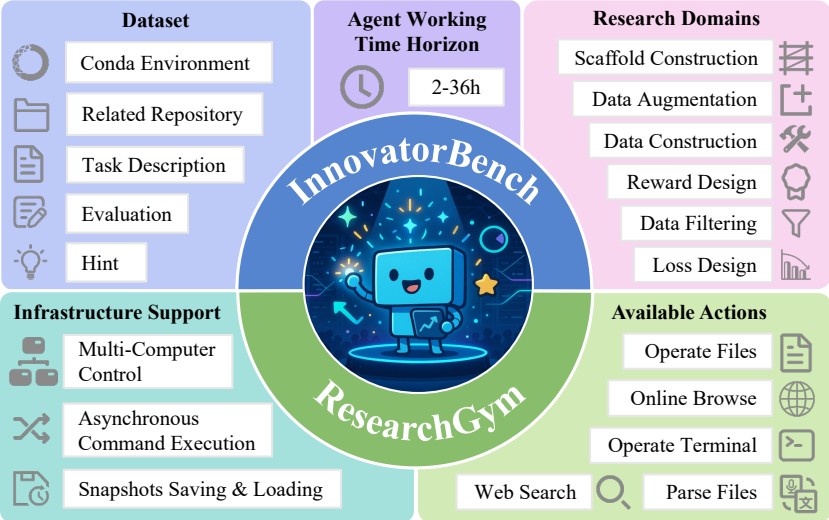

Figure 1: **Overview of InnovatorBench and ResearchGym.** InnovatorBench consists of 20 LLM research tasks from 6 research domains. Each task requires 2-36 hours to complete. ResearchGym provides the infrastructure support for the agent to work on InnovatorBench.

---

*Equal contribution. The order of co-first authors alternates between InnovatorBench and Apollo (https://arxiv.org/pdf/2510.27630). Emails: yunze.wu@sjtu.edu.cn, fdy@bupt.edu.cn, Github Page: https://github.com/GAIR-NLP/InnovatorBench
†Corresponding authors. Email: stefanpengfei@gmail.com

# 1 INTRODUCTION

Artificial intelligence (AI) is becoming central to scientific discovery (Chen et al., 2024; Starace et al., 2025). Traditional workflows require humans to hypothesize, design experiments, implement and debug code, process data, manage resources, and analyze results. As AI rapidly advances, the potential to automate entire research workflows is on the horizon (Liu et al., 2025). We refer to these systems as "AI researchers": agents that integrate multiple stages of research and target human-level behaviors, including insight generation and implementation (Team et al., 2025b). Since Large Language Models (LLMs) act as the "brains" of such agents (Xi et al., 2025), they can finish auxiliary tasks such as data cleaning, augmentation, loss design, reward design, or architecture design as LLM capabilities improve in planning, code generation, and debugging. Moreover, better LLMs agents can propose hypotheses and execute experiments more reliably, which accelerates discovery and feeds back into improving themselves (Liu et al., 2025). As a result, transferring improvements in LLM capabilities into genuine progress for AI research agents requires more than isolated skills. The key question is whether these abilities can be orchestrated into coherent, end-to-end research workflows (Chen et al., 2024; Edwards et al., 2025). This motivates systematic and realistic evaluation of AI research agents and has prompted recent efforts to benchmark them.

Recent efforts to AI research benchmark have provided valuable insights and represent important first steps toward formalizing this emerging area (Starace et al., 2025; Chen et al., 2024; Edwards et al., 2025; Xu et al., 2025; Team, 2025). These studies show that current agents can already achieve non-trivial performance on experiment design, implementation correctness, and even limited replication of advanced research results, establishing clear baselines for progress. At the same time, these benchmarks highlight several structural limitations. Many existing tasks concentrate narrowly on a single performance dimension, such as code implementation accuracy, or parameter tuning (Hua et al., 2025), rather than evaluating the entire research process end to end. Success is often framed as reproducing existing results (Starace et al., 2025), which measures fidelity but **not the capacity for innovation**, new objective design, or architectural creativity. Moreover, the research environments where agents are evaluated are simplified and resource-constrained, so **large-scale and long-horizon training or inference are typically unsupported**, and asynchronous monitoring of processes that span multiple hours is rare (Kon et al., 2025). **Action spaces are also constrained**, preventing agents from engaging in realistic research behaviors such as managing files, executing commands, or browsing literature (Chen et al., 2024). These limitations collectively restrict the conclusions that can be drawn about an agent's potential as a true research collaborator.

In this paper, we try to address these challenges by introducing **InnovatorBench** and a new experimental platform **ResearchGym** that evaluates AI research agents in real scientific practice.

InnovatorBench systematically evaluates core subproblems in LLM research, encompassing data construction (DC), data filtering (DF), and data augmentation (DA), loss design (LD), reward design (RD), and scaffold construction (SC). It consists of 20 tasks, each task isolates a distinct research domain, requiring agents to propose creative methods, implement their own ideas, refine ideas & implementation based on the results, produce concrete outputs, and submit their outputs for several times. To establish baselines and ensure reproducibility, we construct reference solutions and relative evaluation scripts. The reference solutions remain hidden during evaluation so that agents must rely on their own reasoning and design choices. The evaluation scripts quantify metrics like correctness, quality, and/or even the output uncertainty like the entropy of predictions after Reinforcement Learning (RL) (Yu et al., 2025), thereby providing a multifaceted view of agent capabilities. This setup emphasizes both diversity and openness because the tasks span different types of challenges, allow multiple solution strategies, and reward innovation rather than simple replication. Consequently, InnovatorBench moves beyond narrow tests of implementation fidelity and provides a rigorous framework for assessing whether agents can execute end-to-end research workflows that mirror the demands of real LLM development.

In parallel, ResearchGym offers a scalable and realistic environment that addresses limitations of existing platforms (Nathani et al., 2025; Wang et al., 2024a). It provides a rich action space that covers terminal commands, file operations, web search, and web browsing. Building on this foundation, ResearchGym supports large-scale experiments that may run for many hours or even days, with facilities for launching, monitoring, and adapting long-running processes, as well as distributed training across multiple machines and GPUs. It also provides snapshot saving and loading for paus-

Table 1: **Comparison of AI benchmarks across key evaluation dimensions.** *Time Horizon* refers to the time the ReAct-based Agent takes to reach its best score. ML-Bench doesn't report this result.

| Benchmark | Task Resource | Max Eval times | Multi-GPU / Multi-Node | Save and Restore | Creativity | Time Horizon |
|---|---|---|---|---|---|---|
| SWE-bench (Jimenez et al., 2024) | GitHub Issues | 1 | ✗ | ✗ | ✗ | 30m-2h |
| ScienceAgentBench (Chen et al., 2024) | Scientific Papers | 1 | ✗ | ✗ | ✓ | 10m |
| RExBench (Edwards et al., 2025) | NeurIPS, ACL*, etc. Paper | 3 | ✗ | ✗ | ✗ | 6h-12h |
| RE-Bench (Wijk et al., 2024) | Design Manually | 1 | ✗ | ✗ | ✗ | 12m-2h |
| EXP-Bench (Kon et al., 2025) | NeurIPS, ICLR Papers | 1 | ✗ | ✗ | ✗ | 35m |
| PaperBench (Starace et al., 2025) | ICML 2024 Papers | 1 | ✗ | ✗ | ✗ | 1h-3h |
| ML-Bench (Tang et al., 2023) | Kaggle Competitions | 1 | ✗ | ✗ | ✗ | Unknown |
| MLE-bench (Chan et al., 2024) | Kaggle ML Tasks | ∞ | ✗ | ✗ | ✓ | 10m |
| InnovatorBench | NeurIPS, ICLR, etc. Papers | 4 | ✓ | ✓ | ✓ | 2h-36h |

ing and resuming experiments without loss of progress. Importantly, ResearchGym is not tied to a single benchmark; it is a general and extensible platform to which the community can contribute new tasks, datasets, and evaluation protocols, similar to how models and datasets are shared in the HuggingFace (Wolf et al., 2019). This openness allows ResearchGym to evolve with research needs, serving as the foundation for InnovatorBench and as an independent environment for testing new ideas, building baselines, and comparing agents across diverse experimental settings.

To demonstrate the utility of our framework, we deploy a ReAct-based agent on InnovatorBench with several frontier LLMs, including Claude Sonnet 4 (Anthropic, 2025), GPT-5 (OpenAI, 2025), and GLM-4.5 (Zeng et al., 2025), Kimi-K2 (Team et al., 2025a). These experiments provide a systematic basis to analyze how different foundation models perform across diverse subproblems in LLM research, revealing that these models have the potential to handle code-based research tasks longer than 10 hours. However, they **struggle with fragile algorithm design and long-horizon decision making**, often exhibiting impatience, resource mismanagement, poor library choice, and reliance on template-based reasoning. Such comparative analysis offers new insights into the alignment between model capabilities and the requirements of end-to-end agentic research.

Our contributions can be summarized as follows:

• We introduce **InnovatorBench**, the first benchmark to systematically evaluate AI research agents on end-to-end LLM research tasks, spanning data construction, filtering, and augmentation, loss design, reward design, and scaffold construction under multiple dimensions.

• We develop **ResearchGym**, a general and extensible research environment supporting long-duration and distributed experiments, asynchronous execution, snapshot saving and loading, and a broad action space for realistic research workflows.

• We perform an empirical analysis of InnovatorBench across multiple leading LLMs, demonstrating its potential and weaknesses in handling real LLM research tasks.

## 2 RELATED WORK

Recent years have seen growing efforts in developing code agents, which generally fall into two categories: repository-level code benchmarks for assessing specific technical competencies, and agent frameworks that offer execution environments and scaffolding for interactive or long-horizon tasks. Table 1 presents a comparison of several related benchmarks.

**Repository-level code benchmarks.** Several benchmarks focus on assessing whether agents can solve *software engineering* or *machine learning* tasks within realistic repositories. SWE-bench (Jimenez et al., 2024; Yang et al., 2025b;c) and its variants evaluate an agent's ability to resolve GitHub issues by generating executable patches that pass unit tests (Yang et al., 2024; Yao et al., 2023). ScienceAgentBench (Chen et al., 2024) extends this paradigm to scientific domains, requiring agents to write programs that replicate or analyze results derived from real papers. RExBench (Edwards et al., 2025) and EXP-Bench (Kon et al., 2025) target reproducibility and experiment execution, testing whether agents can reconstruct pipelines to reproduce known results. PaperBench

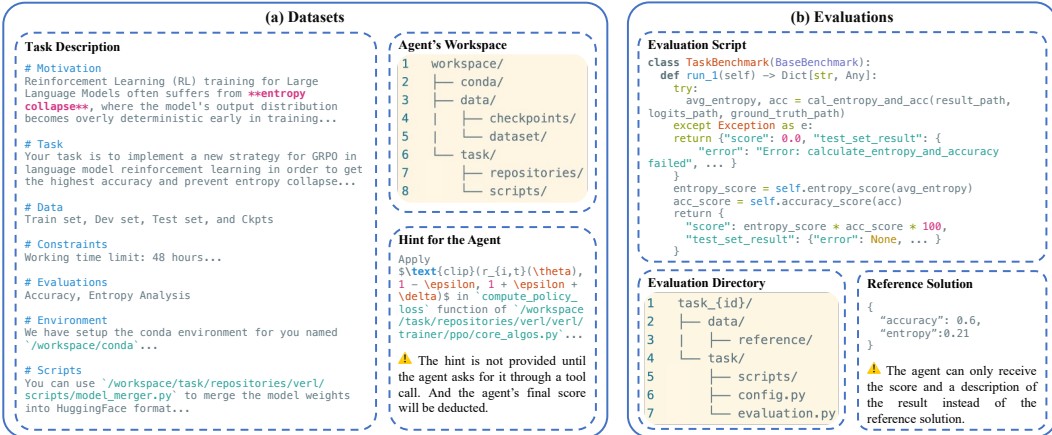

Figure 2: **An illustrative LLM research task from DAPO (Yu et al., 2025). (a) Datasets.** The agent receives a task description and a starter workspace; an optional hint is only revealed upon the agent's explicit request via the `view_hint` tool at a final score penalty. **(b) Evaluations.** An evaluation directory includes evaluation scripts and reference data. Evaluation is performed externally using hidden scripts and reference data. The agent submits its output via the `eval` tool and receives a score with feedback, preventing answer hacking. The full example is in Appendix D.

(Starace et al., 2025) collects machine learning tasks from papers to evaluate large-scale replicability. DatasetResearch (Li et al., 2025) emphasizes dataset discovery and reasoning about data usage. Whereas existing benchmarks focus on narrow aspects of research (e.g., code modification, experiment reproduction), InnovatorBench targets a broader set of LLM-centric research abilities, evaluating agents' proficiency across the entire research lifecycle.

**Agent scaffold and environments.** A complementary line of work focuses on platforms for deploying code-capable agents in interactive environments. OpenHands (Wang et al., 2024a) allows agents to interact with a sandboxed environment via coding, command-line operations, and web browsing. Commercial systems such as Claude Code demonstrate practical coding assistance but prioritize short-term tasks over long-running, research-oriented workflows. Other research systems, including WorldCoder (Tang et al., 2024) and multimodal variants such as OpenHands-Versa (Soni et al., 2025), highlight the potential of tool-augmented agents for general problem solving. Correspondingly, environments like MLGym (Nathani et al., 2025) provide structured contexts for ML-related tasks but often constrain the experiment duration, scale, or action space. A common limitation across these frameworks is the lack of support for extended scientific research: they rarely provide distributed training, asynchronous monitoring of multi-hour jobs, snapshot saving, and integration of open-ended research goals. Our ResearchGym directly addresses these gaps by exposing a rich and extensible action space, enabling long-horizon and distributed experiments, and offering a foundation where new tasks and evaluation protocols can be shared and extended by the community.

# 3 INNOVATORBENCH

InnovatorBench evaluates AI agents' ability to complete end-to-end, innovation-oriented AI research tasks. Each task is derived from an influential AI research paper and its open-source codebase. This coupling captures the full scientific workflow by linking high-level research questions to concrete implementations. As shown in Figure 2, each task entry comprises a task description, an initial starter workspace, a hint for the agent, evaluation scripts, and a reference solution derived from the original research artifacts. The agent's objective is to extensively explore this task in our environment and aim to achieve a performance that surpasses the ground-truth solution.

**Benchmark Overview and Statistics.** InnovatorBench currently comprises 20 research tasks drawn from 14 influential papers, as detailed in Appendix A. These tasks span diverse LLM research areas, including data construction, filtering, and augmentation, loss design, reward design, and scaffold construction. They are sourced from top-tier venues, namely NeurIPS, ICLR, COLM, EMNLP, and

ACL, and the latest publications. This breadth ensures coverage of diverse experimental paradigms, coding practices, and research challenges prevalent in LLM research. Together, these features make InnovatorBench a comprehensive testbed for assessing the capabilities of AI agents.

**Task Description.** Each task description provides the agent with the following components:

(1) *Motivation*: The research motivation and provenance of the question

(2) *Task*: A high-level description of the objective for the agent. To encourage exploration and avoid overfitting to prescribed procedures, we do *not* specify step-by-step instructions; instead, the agent is expected to aim surpasses the reference solution no matter what method it selects.

(3) *Data*: Details of the relevant datasets and checkpoints, including content description, storage paths, file formats, and illustrative examples.

(4) *Constraints*: The operational constraints under which the agent must complete the task, like working time limits, GPU quotas, and output file format.

(5) *Evaluations*: The evaluation metrics like accuracy, F1, and BLEU, and an introduction to the scoring function in this task.

(6) *Environment*: Information about the execution environment, including the conda environment and the workspace directory layout.

(7) *Scripts*: The description of several supplementary unified scripts and repositories that the agent can use in their process.

**Workspace.** The workspace is a writable directory containing essential task artifacts, over which the agent has complete control. The workspace comprises three major components:

(1) *Conda environment*: We pre-build a minimal conda environment that replicates the original paper's setup to run baseline experiments. We recommend not modifying this base environment; however, to preserve the agent's autonomy, we do not prohibit modifying packages when necessary.

(2) *Data*: For the agent to validate whether its proposed methods enhance model performance, we supply complete datasets (training, validation, and a test set without ground-truth for agents' submissions) and model checkpoints suitable for fine-tuning. The agent may also search for, download, and reformat additional data to meet the repository's requirements, or even synthesize new datasets using the provided models or by generating chain-of-thought-style data for augmentation.

(3) *Task*: This directory contains the code repository and a set of helper scripts. The repository is adapted from the original paper's codebase: we remove the implementation of the paper's key novelty and git commit history while keeping the project runnable. In most tasks, the repository is LlamaFactory (Zheng et al., 2024) or Verl (Sheng et al., 2024). The scripts folder offers scripts for data construction, training, inference, and evaluation; the agent may add its own scripts and files.

**Hint for the Agent.** To assist with these challenging tasks, we provide an optional hint for each task. Hints are not included in the workspace; an agent may query their contents via the `view_hint` tool, choosing whether to adopt them. Our main evaluation disabled this tool, while the ablation study provides a hint immediately after the task description.

**Evaluations.** Our evaluation follows a Kaggle-style[1] procedure with multiple submission opportunities and immediate score feedback on the test set. First, a submission is checked for format validity, with failures receiving a score of 0 and an error message. Subsequently, valid submissions are scored based on a function calibrated between a *baseline* (anchored near 0) and a *reference solution* (anchored near 80). The entire evaluation runs externally to the workspace. As shown in Figure 3, in real-time operation, the agent can use the *eval* action to evaluate its results and obtain evaluation details (e.g., results) from the judge system. If the agent evaluates for *max eval times*, the system will automatically use the finish action and stop this process.

**InnovatorBench Construction** We first design 20 raw tasks based on the following principle:

(1) The task can be reapplied; the result is aligned with the original paper.

(2) The task result can gain significant improvement in 2 days

---

[1]https://www.kaggle.com/

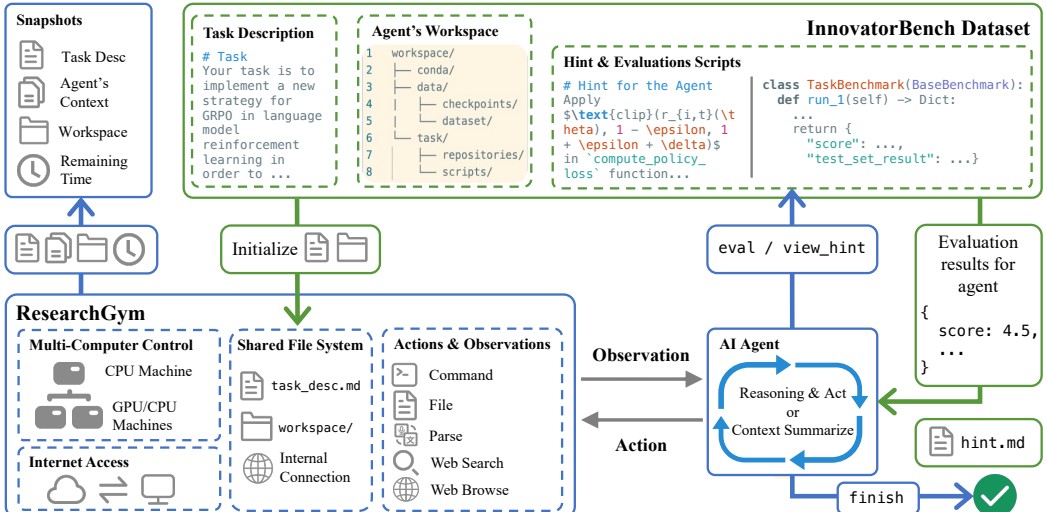

Figure 3: **InnovatorBench evaluates AI agents on research tasks extracted from AI papers.** ResearchGym is initialized with the InnovatorBench dataset; the agent receives a task description and workspace, reasons over observations, and issues tool calls that are translated into actions executed on a target computer, with results returned as structured, agent-readable observations. The agent iterates this process, optionally using `view_hint` for hints and `eval` for submitting answers, until calling `finish`. ResearchGym then performs a final evaluation and saves a state snapshot.

(3) The tasks can evaluate the different abilities of LLM Agents in LLM Research.

(4) The task uses common models like llama3.1 (Grattafiori et al., 2024), Qwen2.5 (Team, 2024; Guo et al., 2025; Qwen et al., 2025), or Qwen2.5-VL (Bai et al., 2025), etc.

More details about the annotation process can be seen in Appendix B.

# 4 RESEARCHGYM

Prior agent systems such as OpenHands (Wang et al., 2024a) and IterativeAgent (Starace et al., 2025) operate within a single Docker container. They execute commands synchronously, so the next action cannot be chosen until the previous one finishes. This design constrains the scale of experiments and reduces action throughput. To overcome these limitations, we introduce ResearchGym, an environment designed to approximate real-world LLM research. Inspired by OpenHands's design, ResearchGym provides 42 primitive actions that agents can freely compose, supports control of multiple machines and asynchronous command execution, and allows users to save and restore environment snapshots. It should also be emphasized that the ResearchGym is not solely used for InnovatorBench, we believe it can be used in other research tasks.

**Actions and Observations.** Actions of ResearchGym are grouped into five families: Command, File, Parse, Web Search, and Web Browse. *Command* actions can manage execution sessions, run commands within a session, and retrieve outputs. *File* actions can perform file operations (e.g., create, edit, delete, read, and search), and query file metadata. *Parse* actions can extract and preview content from multi-modal sources (e.g., images, audio, and video) for text-only models. *Web Search* and *Web Browse* grant networked retrieval and browsing for accessing up-to-date methods and datasets. Each action family is paired with an observation that normalizes raw outputs into a structured, agent-readable return. Details can be referred to Appendix F.

**Multi-Computer Control.** ResearchGym allows agents to control multiple machines (or Docker containers) concurrently via HTTP. Each computer runs an HTTP server to receive and execute terminal commands, allowing an agent initialized on a single machine to orchestrate long-horizon, distributed experiments across a cluster.

**Asynchronous Command Execution.** ResearchGym decouples action execution from selection to prevent decision blocking. Agents can bind commands to specific sessions, or let ResearchGym create new ones. This ensures ongoing jobs continue uninterrupted and enable immediate subsequent planning. Agents can later retrieve the result via `get_session_output` asynchronously. To avoid nonsensical actions during model training, ResearchGym provides a dedicated `sleep` action.

**Snapshots Saving and Loading.** A snapshot records the task specification, the agent's context, the final state of the workspace, and the remaining time budget. ResearchGym can periodically save the full state as snapshots, and it can restore the system from any snapshot. Snapshots support branching. Experiments can resume from different points or proceed along multiple branches.

**Pipeline.** Figure 3 depicts the end-to-end interaction loop. The process begins when ResearchGym loads a task from InnovatorBench, providing the agent with a task description as its initial observation and a starter workspace. Given an observation, the agent reasons and issues a tool call. If it's not a command action, the ResearchGym will produce it locally; otherwise, ResearchGym converts this call into an action, wraps it in an HTTP request, and dispatches it to a target machine. The target machine executes the action or launches it as a background process. ResearchGym packages the outcome as a new observation in an agent-readable format. Synchronous actions immediately update the workspace, whereas asynchronous actions return a session ID and status for the agent to poll with subsequent commands. The agent repeats this loop, optionally submitting answers for evaluation and consulting hints when needed. When the agent deems the task complete, it invokes `finish`. ResearchGym performs a final evaluation, saves a snapshot, and finalizes the task.

## 5 EXPERIMENTS AND RESULTS

### 5.1 EXPERIMENTAL SETUP

We evaluate leading LLMs commonly used in related benchmarks on InnovatorBench. Specifically, we consider Claude Sonnet 4 (Anthropic, 2025), GPT-5 (OpenAI, 2025), and GLM-4.5 (Zeng et al., 2025), Kimi-K2 (Team et al., 2025a), Qwen3-32b (Yang et al., 2025a) using a ReAct-style agent (Yao et al., 2023). The agent has the fundamental thought and action capabilities, augmented by a summarization capability. When the context length nears the model's maximum, the agent will summarize the earlier half of the context. All models are wrapped as agents and executed inside a Docker container on Ubuntu 22.04 with 800 GB of memory. The agent can also, via a cluster HTTP service, dispatch additional compute to server(s) with 8x 80 GB GPUs and 1600 GB of memory each, with the number of servers allocated varying by task. We also provide a clean working directory containing the relevant data, a starter code repository, and the task description for each task. Data Construction and Data Augmentation can connect the internet. We disable the web search and browse tools in other tasks.

Table 2: **Performance comparison on various LLMs when tested against various research domains.** *Final Score*: last submission score; *Best Score*: highest achieved score. Details of all research tasks can be referred to Appendix C.

| Research Domain | Claude Sonnet 4 | | GPT-5 | | GLM-4.5 | | Kimi-K2 | | Qwen3-32b | |
|---|---|---|---|---|---|---|---|---|---|---|
| | Final Score | Best Score | Final Score | Best Score | Final Score | Best Score | Final Score | Best Score | Final Score | Best Score |
| Data Construction | **25.47** | **26.88** | 8.41 | 8.41 | 15.29 | 22.65 | 14.01 | 14.08 | 0.00 | 0.00 |
| Data Filtering | **30.89** | **31.47** | 8.97 | 9.48 | 5.16 | 5.36 | 7.39 | 7.97 | 0.00 | 0.00 |
| Data Augmentation | 22.73 | 22.73 | 0.00 | 0.00 | **25.49** | **25.49** | 2.47 | 2.47 | 0.00 | 0.00 |
| Loss Design | **12.98** | **12.98** | 0.04 | 2.74 | 7.63 | 7.63 | 0.00 | 0.00 | 0.00 | 0.00 |
| Reward Design | **11.56** | **11.56** | 0.00 | 0.00 | 0.00 | 0.00 | 3.23 | 3.23 | 0.00 | 0.00 |
| Scaffold Construction | 36.63 | 37.74 | **60.07** | **60.07** | 3.33 | 3.33 | 3.33 | 3.33 | 0.00 | 0.00 |
| Weighted Average | **24.01** | **24.54** | 12.04 | 12.52 | 11.85 | 13.35 | 5.35 | 5.45 | 0.00 | 0.00 |

### 5.2 MAIN RESULTS AND FINDINGS

As demonstrated in Table 2, we compare three agents across six research-oriented tasks and report both the final and best scores achieved. Overall, all the agents get non-zero scores, which show they **have the potential to handle code-based research tasks**. *Claude Sonnet 4* demonstrates the most superior performance among its counterparts, attaining the highest average final score and best

score, and leading on four of six tasks. *GPT-5* and *GLM-4.5* yield middling results on final score and best score, respectively. *Qwen3-32b* gains 0, because its context window is too small and often omits important information when summarizing. Besides, we also obtain the following findings:

**All LLMs have relatively higher scores on data-related tasks than on algorithm-related tasks.** This difference arises from the nature of these tasks, tasks such as data construction, filtering, and augmentation are inherently more robust: it is relatively tolerant of minor noise. For example, the agent can gain a relatively high score in data construction as long as it find the data with the same topic. In contrast, algorithmic design tends to be more brittle; imperfect reward or loss functions can lead to catastrophic failures like gradient explosion or systematically flawed policies.

**It is hard for models to use appropriate tools in algorithm-related tasks.** We discover that Claude Sonnet 4 performs relatively better than other LLMs on loss/reward design, primarily due to its reliable tool use. Trace inspection reveals that GPT-5 enters a high-frequency loop once training begins, causing early termination, while GLM-4.5 wrongly specifies critical tool parameters sometimes and stalls before training starts. Kimi-K2 cannot generate correct code in most cases. However, Claude Sonnet 4 consistently produces executable code and correctly suspends activity during training without intervention. These findings suggest that reliability in tool-grounded execution is the key determinant of success in loss/reward design tasks.

**GPT-5's code is more robust in Scaffold Construction.** GPT-5 excels notably in scaffold construction, achieving a score of 60.07, which raises its overall average to 12.04. Analysis shows its generated scaffolds are most robust, attributable to three key design choices: explicitly restating the options provided in the prompt to prevent invalid selections, allowing up to three retries instead of immediately resorting to a fallback answer upon timeout, and enforcing a strict output format to reduce evaluation failures caused by formatting issues.

## 5.3 Performance of Model with Ground Truth Hint

Table 3: **Effect of hint provision on agent performance across research domains.** Comparison between Claude Sonnet 4 with and without hints. *Final Score*: last submission score; *Best Score*: highest achieved score; *Execution Time*: agent runtime (hours); *Cost*: monetary expenditure (USD).

| Research Domain | Claude Sonnet 4 w/ Hint | | | | Claude Sonnet 4 | | | |
|---|---|---|---|---|---|---|---|---|
| | Final Score | Best Score | Execution Time | Cost | Final Score | Best Score | Execution Time | Cost |
| Data Construction | 15.21 | 19.80 | 1.78 | 25.56 | **25.47** | **26.87** | 3.24 | 33.09 |
| Data Filtering | 16.87 | 20.02 | 6.97 | 32.15 | **30.89** | **31.47** | 4.80 | 32.57 |
| Data Augmentation | 1.00 | 1.00 | 3.71 | 26.03 | **22.73** | **22.73** | 4.48 | 30.70 |
| Loss Design | **22.65** | **25.32** | 9.05 | 45.11 | 12.98 | 12.98 | 6.32 | 34.78 |
| Reward Design | **15.06** | **15.06** | 6.22 | 41.69 | 11.56 | 11.56 | 9.23 | 46.14 |
| Scaffold Construction | 21.04 | 27.71 | 3.73 | 23.22 | **36.63** | **37.74** | 6.43 | 28.65 |
| Weighted Average | 13.88 | 16.67 | 4.87 | 30.86 | **24.01** | **24.54** | 5.13 | 32.92 |

Table 3 compares model performance with and without ground-truth hints. Hints substantially improve performance in Loss Design and Reward Design. These domains are **inherently more exploratory in nature**, requiring the agent to devise novel solutions based on test data and algorithmic understanding. With the solution provided, the agent shifts from exploration to implementation, focusing on replicating a known approach rather than inventing one, thereby increasing success rates.

In contrast, performance declines in Data Construction, Filtering, and Augmentation when hints are supplied. Although the ground truth is provided, the model's **coding proficiency becomes a bottleneck** as mentioned in §5.4. The model's tendency to rigidly replicate the hint introduces implementation errors—particularly in script generation—where minor mismatches critically undermine functionality. Consequently, the presence of the hint actually degrades performance compared to the symbolic method used by the agent when no hint is provided.

These results underscore that strong performance in research-oriented tasks demands both creativity and code implementation ability of the agent. A deficiency in either high-level reasoning or low-level coding significantly limits overall effectiveness.

## 5.4 CASE STUDY

**(a) Impatience**

The training costs **10** hours. The agent still has **21** hours.

**Observation**:
```
(pid=2337004) Running step 1:  17%|███        | 1.00/6.00
[03:31<17:36, 211s/it]
...
(pid=2348711) Compute log probs 2:  45%|███████   | 9.00/20.0
[01:12<00:40, 3.70s/it]
```
**Think**:
The second training is progressing very slow, Given that I have limited time and only 3 eval submissions, I should consider trying a different approach. Let me check if there's been any progress with checkpoints saved and consider trying a third, simpler approach that might train faster.
**Action**:
```
{"name": "kill_session_processes", "arguments": {"session_id":
"gpu_train", "computer_ip": "<a_real_ip>"}}
```

**(b) Resource Mismanagement**

**Action 174:**
```
{"name": "run_command", "arguments": {"session_id": "train",
"computer_ip": "<gpu_1's_real_ip>", "command": "cd /workspace/task/scripts
&& serve run llm:build_app model=/workspace/data/checkpoints/DeepSeek-R1-
DualAlign tensor-parallel-size=1"}}
(After several steps, the inference is still running)
```
**Action 229:**
```
{"name": "run_command", "arguments": {"session_id": "eval", "computer_ip":
"<gpu_1's_real_ip>", "command": "cd /workspace/task/repositories/LLaMA-
Factory && bash /workspace/task/scripts/train.sh"}}
```
**Observation 231 (observation of GetSessionOutputAction):**
```
...
torch.OutOfMemoryError: CUDA out of memory. Tried to allocate 892.00
MiB. GPU 0 has a total capacity of 79.19 GiB of which 775.94 MiB.
```

**(c) Selection of Suboptimal Library**

**Action (create file) :**
```
from transformers import AutoTokenizer, AutoModelForCausalLM
class QwenAssistedCleaner:
    ...
    self.model = AutoModelForCausalLM.from_pretrained(
        self.model_path, torch_dtype=torch.float16,
        device_map="auto", trust_remote_code=True)
    ...
    inputs = self.tokenizer(prompt, return_tensors="pt",
        truncation=True, max_length=1024)
    with torch.no_grad():
        outputs = self.model.generate(
            inputs.input_ids, max_new_tokens=512, do_sample=True,
            temperature=0.1, pad_token_id=self.tokenizer.eos_token_id)
    ...
```

**(d) Template-based Reasoning**

**Action (create file) :**
```
def create_enhanced_training_data(original_data, output_path):
    reasoning_templates = ["Let me analyze this step by step:\n\n1. First,
I'll identify the key information given in the problem.\n2. Next, I'll
determine what scientific principles or concepts apply.\n3. Then, I'll ...
After careful analysis, the answer is {answer}.",...]
    for item in original_data:
        ...
        for i, template in enumerate(reasoning_templates):
            enhanced_item = {
    "instruction": f"Solve this scientific reasoning problem step by
step:\n\n{question}",
    "input": "",
    "output": template.format(answer=answer)}
            training_data.append(enhanced_item)
    ...
```

Figure 4: **Four representative cases of agents' actual failures.**

**Impatience.** As shown in Figure 4(a), the training run takes about 10 hours; at that point, the agent knows it still has roughly 21 hours of budget. It is sufficient to wait for completion rather than terminating the process. However, the agent wants to find a more efficient way to train the model and kill the training process, which causes sub-optimal result. The *objective mis-specification and shortsighted decision-making* reflect the agent's impatience.

**Resource Mismanagement.** Figure 4(b) demonstrates that the agent first launches an inference script with one GPU; 55 steps later, on the same computer, it launches a training script that requires all GPUs, causing resource contention. The agent no longer finds that an inference job was already active after more than 50 steps, shows the LLM's weakness in *degraded memory and attention*

**Selection of Suboptimal Library.** Figure 4(c) depicts that the agent systematically opts for scale-mismatched implementations: it continues to run inference with Transformers in high-throughput settings instead of adopting the more efficient vLLM. This is because the *time-budget constraint does not provide a direct, learnable feedback signal* that rewards efficiency, so it fails to shape the agent's decisions. It may also be attributed to the *lack of training data* for the optimal library, like vLLM, since the optimal library is relatively new.

**Template-based Reasoning.** Figure 4(d) shows that when synthesizing chain-of-thought (CoT) rationales for QA data augmentation, the agent often instantiates a highly templated, semantically vacuous reasoning pattern and batch-concatenates the question and answer, rather than reasoning from the problem's actual semantics. We find that this pattern often appears after the agent fails to generate a correct CoT via VLLM. The agents can't figure out why it needs to synthesize CoT and just do it mechanically. Although this case shows the *agentic ability*, it also reflects the agent's *lack of understanding of high-level intent*.

## 5.5 TEST-TIME SCALING PERFORMANCE

Figure 5 compares test-time scaling between InnovatorBench and PaperBench. The score represents the average score up to that time point. For example, if the agent *eval* its result at 1,5,10,13 hours in task 1 and 2,10,23,24 hours in task 2. The score of *hour 16* will be the average score of *hour 13* in task 1 and *hour 10* in task 2. As shown in the figure, agents achieve their best performance on PaperBench in approximately 1.75 hours, but require over 11 hours on InnovatorBench, indicating its greater complexity. This disparity arises because complex tasks like Data Augmentation and Reward Design involve extended training phases, making them more time-intensive than tasks like

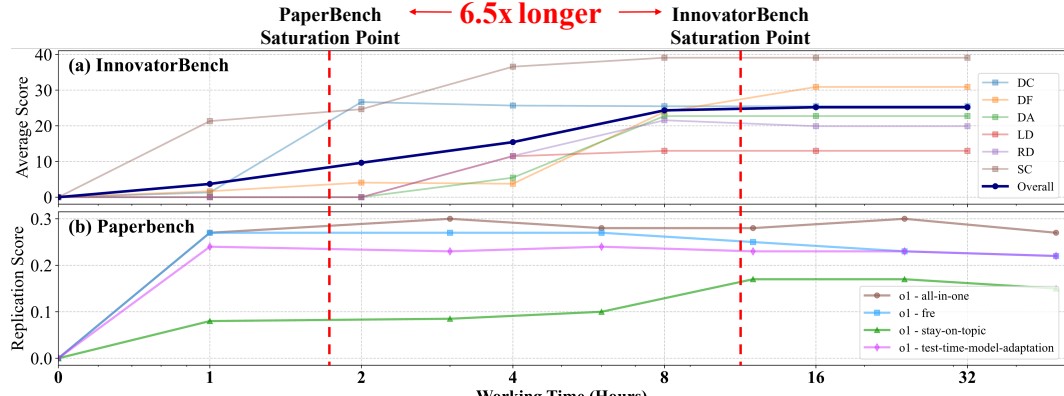

Figure 5: **Test-time scaling: InnovatorBench vs. PaperBench (Starace et al., 2025).** Paper-Bench's result comes from the original paper. Agents require about $6.5\times$ longer test-time to reach the saturation point on InnovatorBench, highlighting that our benchmark's difficulty stems from the need for extended runtime before performance plateaus.

Data Construction. This trend shows that as task complexity increases, environment interaction costs dominate the overall working time. Since the time costs can reflect the difficulty of the tasks, we believe that InnovatorBench is more difficult than PaperBench and will be the next generation of the code-based research benchmark.

# 6 CONCLUSION

In conclusion, this work introduces two key contributions to the development of AI research agents: InnovatorBench, a comprehensive benchmark for evaluating end-to-end LLM research tasks, and ResearchGym, an extensible platform that supports large-scale, long-horizon experiments and realistic research workflows. InnovatorBench goes beyond basic task reimplementation, offering a rigorous framework that evaluates agents' ability to address complex LLM research challenges across multiple dimensions. This emphasis on innovation, adaptability, and creative problem-solving ensures a more comprehensive assessment of AI research agents. Empirical results using leading LLMs reveal promising capabilities in code-based tasks, but also expose weaknesses in reward design, resource management, and long-horizon planning. Together, these contributions provide a foundation for rigorous, real-world evaluation of AI agents, supporting their development as effective tools for scientific discovery.

## ETHICS STATEMENT

This work adheres to the ICLR Code of Ethics. In this study, no human subjects or animal experimentation were involved. All datasets used, including the InnovatorBench we proposed, were sourced in compliance with relevant usage guidelines, ensuring no violation of privacy. We have taken care to avoid any biases or discriminatory outcomes in our research process. No personally identifiable information was used, and no experiments were conducted that could raise privacy or security concerns. We are committed to maintaining transparency and integrity throughout the research process.

## REPRODUCIBILITY STATEMENT

We have made every effort to ensure that the results presented in this paper are reproducible. The experimental setup, including training steps, model configurations, and hardware details, is described in detail in the paper. We have also provided a full description of our InnovatorBench and Research-Gym to assist others in reproducing our experiments. We believe these measures will enable other researchers to reproduce our work and further advance the field.

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

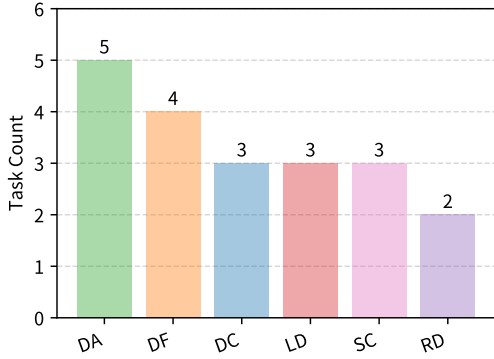

Figure 6: InnovatorBench's dataset comprises tasks from a diverse set of AI research categories. *DA* denotes Data Augmentation, *DC* stands for Data Construction, *DF* represents Data Filtering, *LD* is the Loss Design, *SC* denotes Scaffold Construction, and *RD* means Reward Design.

## A  EXTENDED DETAILS OF THE INNOVATORBENCH DATASET

Figure 6 shows the task composition of InnovatorBench, and Table 4 presents the details of each task in InnovatorBench.

Table 4: The introduction of the InnovatorBench

| ID | Paper | Key Description | Constrain | Research Domains |
|---|---|---|---|---|
| 1 | DatasetResearch: Benchmarking Agent Systems for Demand-Driven Dataset Discovery (Li et al., 2025) | Collect or synthesize a politics-domain news summarization dataset consisting of English news articles with corresponding one-sentence human-written summaries for fine-tuning. | Llama-3.1-8B-Instruct dataset discovery / synthesis 48h, 8×80GB GPUs | Data Construction |
| 2 | DatasetResearch: Benchmarking Agent Systems for Demand-Driven Dataset Discovery (Li et al., 2025) | Build a medical English–Tamil parallel dataset and fine-tune Llama-3.1-8B-Instruct for translation. | Llama-3.1-8B-Instruct dataset discovery / synthesis 48h, 8×80GB GPUs | Data Construction |
| 3 | DatasetResearch: Benchmarking Agent Systems for Demand-Driven Dataset Discovery (Li et al., 2025) | Build a 5K–10K real-world document summarization dataset and fine-tune Llama-3.1-8B-Instruct to generate concise, accurate summaries. | Llama-3.1-8B-Instruct dataset discovery / synthesis 48h, 8×80GB GPUs | Data Construction |
| 4 | DatasetResearch: Benchmarking Agent Systems for Demand-Driven Dataset Discovery (Li et al., 2025) | Fine-tune Llama-3.1-8B on medical Q&A data to improve USMLE-style multiple-choice accuracy from 26% baseline to 95% target. | Llama-3.1-8B-Instruct dataset or synthesis 48h, 8×80GB GPUs | Data Construction |
| 5 | Programming Every Example: Lifting Pretraining Data Quality Like Experts at Scale (Zhou et al., 2024) | Design and implement a systematic cleaning pipeline for 100K raw web texts to produce high-quality data for LLM pre-training. | 5h, 8×80GB GPUs high efficiency | Data Filtering |
| 6 | LIMO: Less is More for Reasoning (Ye et al., 2025b) | Select 800 quality math problems from 10K to build a training set for reasoning optimization. | fixed training hyperparameter select 800 problems 48h, 16×80GB GPUs | Data Filtering |
| 7 | How Do Your Code LLMs Perform? Empowering Code Instruction Tuning with High-Quality Data (Wang et al., 2024b) | Filter code instruction datasets by removing contaminated samples and selecting top 160k highest-difficulty problems for clean model training. | 8h, 8×80GB GPUs | Data Filtering |
| 8 | Supergpqa: Scaling LLM Evaluation Across 285 Graduate Disciplines (Du et al., 2025) | Enhance and fine-tune Qwen2.5-7B using enriched multidisciplinary scientific reasoning data to maximize test set accuracy. | 48h, 8×80GB GPUs final model trained from Qwen2.5-7B | Data Augmentation |

*Continued on next page*

| ID | Paper | Key Description | Constrain | Research Domains |
|---|---|---|---|---|
| 9 | NuminaMath: The Largest Public Dataset in AI4Maths with 860k Pairs of Competition Math Problems and Solutions (Li et al., 2024) | Fine-tune Qwen2.5-7B-Instruct on mathematical reasoning problems using dataset enhancement and auxiliary models to maximize test accuracy beyond 25.9% baseline. | 48h, 8×80GB GPUs final model trained from Qwen2.5-7B | Data Augmentation |
| 10 | DeepResearcher: Scaling Deep Research via Reinforcement Learning in Real-world Environments (Zheng et al., 2025) | Synthesize search-enhanced reasoning data with Qwen-2.5-72B and fine-tune Qwen-2.5-7B to maximize test set performance. | 24h, 8×80GB GPUs fixed inference script final model trained from Qwen2.5-7B | Data Augmentation |
| 11 | Towards Dynamic Theory of Mind: Evaluating LLM Adaptation to Temporal Evolution of Human States (Xiao et al., 2025) | Synthesize dynamic Theory-of-Mind training data and fine-tune Qwen2-7B-Instruct to predict evolving mental states in multi-step social scenarios. | 48h, 8×80GB GPUs Qwen2-7B-Instruct / SFT only | Data Augmentation |
| 12 | MAC: A Live Benchmark for Multimodal Large Language Models in Scientific Understanding (Jiang et al., 2025) | Augment scientific image-text datasets and fine-tune Qwen2.5-VL-7B-Instruct to improve multimodal reasoning for journal cover visual understanding. | 24h, 8×80GB GPUs final model trained from Qwen2.5-VL-7B-Instruct | Data Augmentation |
| 13 | Flexible Realignment of Language Models (Zhu et al., 2025) | Implement a DualAlign algorithm in LLaMA-Factory to efficiently realign DeepSeek-R1-Distilled-Qwen-1.5B with DeepScaleR-Preview-1.5B for improved reasoning efficiency. | fixed training hyperparameter 48h, 8×80GB GPUs | Loss Design |
| 14 | DAPO: An Open-Source LLM Reinforcement Learning System at Scale (Yu et al., 2025) | Implement a GRPO variant in Verl framework to prevent entropy collapse during RL training while maximizing accuracy on mathematical reasoning tasks. | max 24h per training 48h, 8×80GB GPUs Special output format | Loss Design |
| 15 | Robust Preference Optimization via Dynamic Target Margins (Sun et al., 2025) | Develop a GammaPO algorithm that adaptively adjusts reward margins based on preference clarity to outperform SimPO on AlpacaEval2 benchmarks. | 24h, 8×80GB GPUs Modify the training script only | Loss Design |
| 16 | Search-R1: Training LLMs to Reason and Leverage Search Engines with Reinforcement Learning (Jin et al., 2025) | Design and implement a reward function in the Verl framework to train Qwen-2.5-3B for search-augmented reasoning tasks with exact match evaluation. | 48h, 8×80GB GPUs final model trained from Qwen2.5-3B | Reward Design |
| 17 | GUI-R1: A Generalist R1-Style Vision-Language Action Model For GUI Agents (Luo et al., 2025) | Implement unified reward function in Verl framework to train GUI grounding models for multi-platform action prediction exceeding ScreenSpot baseline accuracies. | 24h, 8×80GB GPUs final model trained from Qwen2.5-VL-7B | Reward Design |
| 18 | DeepResearcher: Scaling Deep Research via Reinforcement Learning in Real-world Environments (Hu et al., 2024) | Build a prompt-based deep research agent using GPT-4.1 and web tools to handle complex multi-step research questions with accurate source-attributed answers. | 24h, 0 GPU GPT-4.1 for research GPT-4.1-mini for browsing | Scaffold Construction |
| 19 | Visual SKETCHPAD: Sketching as a Visual Chain of Thought for Multimodal Language Models (Hu et al., 2024) | Develop efficient multimodal mathematical reasoning workflow using GPT-4o to solve geometry, graph connectivity, maxflow, and convexity problems with structured JSON outputs. | 12h, 0 GPU GPT-4o via API | Scaffold Construction |
| 20 | Visual SKETCHPAD: Sketching as a Visual Chain of Thought for Multimodal Language Models (Hu et al., 2024) | Develop efficient visual reasoning system using GPT-4o and visual tools to solve vstar, blink_viscorr, blink_jigsaw, and blink_depth tasks with structured JSON outputs. | 12h, 1×24GB GPU GPT-4o via API | Scaffold Construction |

## B DATASET CURATION AND BENCHMARK CONSTRUCTION DETAILS

**InnovatorBench Construction** We first design 20 raw tasks based on the following principle:

(1) The task can be reapplied; the result is aligned with the original paper.

(2) The task result can gain significant improvement in 2 days

(3) The tasks can evaluate the different abilities of LLM Agents in LLM Research.

(4) The task uses common models like llama3.1 (Grattafiori et al., 2024), Qwen2.5 (Team, 2024; Guo et al., 2025; Qwen et al., 2025), or Qwen2.5-VL (Bai et al., 2025), etc.

There are 13 annotators to annotate InnovatorBench. Each task costs from 3 days to 2 weeks for the annotators to construct the workspace and evaluation code. After collecting 20 tasks, 2 authors further organize these tasks, workspaces, and evaluations into ResearchGym. Each annotators were asked to reapply the original paper and gain the reference score, and the baseline score often comes from the base model's result. After obtaining these two scores, the annotators were asked to design the score function based on these scores, usually a linear interpolation. The score function has 2 principles: (1) the baseline score should result in a final score of 0, while if agents gain a score higher than the baseline score, their final score should not be 0, and (2) the reference score should be about 80.

After testing the first version of InnovatorBench, we found that even the most advanced model can't generate and save the SFT data correctly, as mentioned in Figure 4, so we just changed the task a little bit to reduce the difficulty in the Data Argument by adding some relevant scripts.

## C  Detailed Experimental Results

### C.1  Main Results

Table 5: **Performance comparison of each task on various models when tested against various evaluation metrics.** FS denotes Final Score, BS represents Best Score, ET stands for Execution Time in hours, and Cost is the monetary spend in USD.

| Task | Claude Sonnet 4 | | | | GPT-5 | | | | GLM-4.5 | | | | Kimi-K2 | | | |
|---|---|---|---|---|---|---|---|---|---|---|---|---|---|---|---|---|
| | FS | BS | ET | Cost | FS | BS | ET | Cost | FS | BS | ET | Cost | FS | BS | ET | Cost |
| 1 | 19.05 | 19.05 | 1.53 | 27.02 | 0.00 | 0.00 | 2.41 | 16.09 | 20.51 | 20.51 | 0.84 | 1.63 | 0.00 | 0.00 | 1.95 | 4.92 |
| 2 | 39.35 | 39.35 | 4.26 | 47.41 | 0.00 | 0.00 | 0.79 | 4.16 | 13.26 | 13.26 | 0.83 | 1.91 | 0.20 | 0.20 | 3.57 | 5.97 |
| 3 | 17.41 | 18.23 | 4.98 | 30.42 | 0.00 | 0.00 | 1.44 | 7.49 | 18.85 | 18.85 | 2.06 | 4.31 | 14.70 | 14.98 | 4.25 | 4.40 |
| 4 | 26.09 | 30.87 | 2.21 | 27.51 | 33.62 | 33.62 | 3.98 | 19.42 | 8.55 | 37.97 | 1.37 | 5.70 | 41.16 | 41.16 | 2.53 | 3.40 |
| 5 | 5.00 | 5.00 | 0.54 | 6.86 | 13.36 | 14.88 | 0.60 | 2.37 | 5.00 | 5.00 | 0.16 | 0.65 | 12.76 | 13.12 | 0.64 | 1.40 |
| 6 | 81.37 | 81.74 | 11.45 | 65.09 | 0.00 | 0.00 | 1.58 | 4.28 | 0.00 | 0.00 | 5.32 | 11.11 | 5.00 | 5.00 | 3.33 | 6.14 |
| 7 | 6.29 | 7.66 | 2.41 | 25.75 | 13.55 | 13.55 | 1.28 | 3.94 | 10.48 | 11.08 | 0.56 | 3.77 | 4.40 | 5.80 | 1.57 | 2.03 |
| 8 | 27.33 | 27.33 | 4.83 | 21.57 | 0.00 | 0.00 | 6.32 | 36.15 | 37.30 | 37.30 | 1.54 | 4.87 | 7.33 | 7.33 | 3.38 | 3.98 |
| 9 | 0.00 | 0.00 | 7.08 | 36.01 | 0.00 | 0.00 | 2.19 | 4.75 | 0.00 | 0.00 | 0.64 | 3.95 | 0.00 | 0.00 | 11.57 | 7.36 |
| 10 | 0.00 | 0.00 | 2.34 | 41.15 | 0.00 | 0.00 | 1.06 | 3.86 | 0.00 | 0.00 | 0.56 | 3.95 | 0.00 | 0.00 | 0.80 | 0.46 |
| 11 | 86.34 | 86.34 | 5.49 | 32.58 | 0.00 | 0.00 | 3.99 | 4.77 | 5.00 | 5.00 | 2.93 | 7.53 | 5.00 | 5.00 | 6.62 | 6.24 |
| 12 | 0.00 | 0.00 | 2.65 | 22.19 | 0.00 | 0.00 | 0.86 | 2.99 | 85.15 | 85.15 | 8.45 | 8.05 | 0.00 | 0.00 | 1.29 | 4.46 |
| 13 | 0.00 | 0.00 | 4.97 | 58.67 | 0.00 | 0.00 | 1.59 | 12.12 | 0.00 | 0.00 | 3.31 | 10.74 | 0.00 | 0.00 | 7.28 | 10.68 |
| 14 | 4.50 | 4.50 | 10.40 | 26.29 | 0.12 | 8.21 | 16.23 | 80.50 | 0.00 | 0.00 | 0.68 | 5.47 | 0.00 | 0.00 | 4.70 | 12.20 |
| 15 | 34.44 | 34.44 | 3.58 | 19.38 | 0.00 | 0.00 | 2.27 | 6.60 | 22.90 | 22.90 | 4.71 | 18.78 | 0.00 | 0.00 | 3.48 | 8.49 |
| 16 | 0.00 | 0.00 | 8.91 | 51.67 | 0.00 | 0.00 | 1.74 | 8.67 | 0.00 | 0.00 | 0.63 | 3.31 | 0.00 | 0.00 | 1.86 | 6.60 |
| 17 | 23.11 | 23.11 | 5.66 | 32.81 | 0.00 | 0.00 | 2.04 | 9.63 | 0.00 | 0.00 | 0.13 | 1.20 | 6.47 | 6.47 | 3.79 | 6.33 |
| 18 | 16.67 | 20.00 | 14.01 | 44.25 | 0.00 | 0.00 | 1.26 | 6.56 | 0.00 | 0.00 | 1.34 | 5.47 | 0.00 | 0.00 | 2.42 | 1.96 |
| 19 | 63.95 | 63.95 | 2.18 | 23.30 | 92.45 | 92.45 | 1.77 | 3.70 | 10.00 | 10.00 | 0.64 | 3.78 | 10.00 | 10.00 | 5.66 | 2.87 |
| 20 | 29.26 | 29.26 | 3.08 | 18.41 | 87.76 | 87.76 | 5.05 | 36.38 | 0.00 | 0.00 | 0.99 | 3.04 | 0.00 | 0.00 | 0.68 | 3.47 |
| Avg. | **24.01** | **24.54** | **5.13** | **32.92** | 12.04 | 12.52 | 2.92 | 13.72 | 11.85 | 13.35 | 1.92 | 5.68 | 5.35 | 5.45 | 3.57 | 5.17 |

The benchmark evaluated the performance of three large language models — Claude Sonnet 4, GPT-5, GLM-4.5, and Kimi-K2 — on multiple tasks with varying priority levels. For each task, the metrics recorded include the final score, the highest score, and the runtime (in hours). This analysis focuses on comparing model effectiveness (scores) and efficiency (time cost).

## C.2 Performance of Model with Ground Truth Hint

Table 6 presents the performance, execution time, and cost results between Claude Sonnet 4 with the hint and Claude Sonnet 4 without the hint.

Table 6: **Performance comparison between Claude4-hint and Claude4 on evaluation metrics, runtime, and cost.**

| Task | Claude4-hint | | | | Claude4 | | | |
|---|---|---|---|---|---|---|---|---|
| | Final Score | Best Score | Run Time | Cost | Final Score | Best Score | Run Time | Cost |
| 1 | 6.52 | 18.52 | 0.68 | 17.97 | 19.05 | 19.05 | 1.53 | 27.02 |
| 2 | 8.68 | 8.68 | 0.66 | 29.87 | 39.35 | 39.35 | 4.26 | 47.41 |
| 3 | 8.53 | 12.88 | 3.85 | 27.62 | 17.41 | 18.23 | 4.97 | 30.42 |
| 4 | 37.10 | 39.13 | 1.93 | 26.78 | 26.09 | 30.87 | 2.21 | 27.51 |
| 5 | 12.80 | 13.24 | 1.75 | 12.74 | 5.00 | 5.00 | 0.54 | 6.86 |
| 6 | 29.49 | 34.49 | 14.13 | 61.68 | 81.37 | 81.74 | 11.45 | 65.09 |
| 7 | 8.32 | 12.32 | 5.04 | 22.04 | 6.29 | 7.66 | 2.41 | 25.75 |
| 8 | 0.00 | 0.00 | 6.99 | 43.19 | 27.33 | 27.33 | 4.83 | 21.57 |
| 9 | 0.00 | 0.00 | 6.68 | 31.10 | 0.00 | 0.00 | 7.08 | 36.01 |
| 10 | 0.00 | 0.00 | 2.78 | 24.63 | 0.00 | 0.00 | 2.34 | 41.15 |
| 11 | 5.00 | 5.00 | 0.67 | 15.42 | 86.34 | 86.34 | 5.49 | 32.58 |
| 12 | 0.00 | 0.00 | 1.44 | 15.80 | 0.00 | 0.00 | 2.65 | 22.19 |
| 13 | 0.00 | 0.00 | 3.33 | 65.13 | 0.00 | 0.00 | 4.97 | 58.67 |
| 14 | 29.37 | 37.39 | 18.34 | 34.88 | 4.50 | 4.50 | 10.40 | 26.29 |
| 15 | 38.57 | 38.57 | 5.49 | 35.32 | 34.44 | 34.44 | 3.58 | 19.38 |
| 16 | 0.00 | 0.00 | 4.71 | 43.57 | 0.00 | 0.00 | 8.91 | 51.67 |
| 17 | 30.13 | 30.13 | 7.73 | 39.81 | 23.11 | 23.11 | 5.66 | 32.81 |
| 18 | 0.00 | 20.00 | 7.48 | 35.77 | 16.67 | 20.00 | 14.01 | 44.25 |
| 19 | 53.12 | 53.12 | 0.54 | 13.65 | 63.95 | 63.95 | 2.18 | 23.30 |
| 20 | 10.00 | 10.00 | 3.19 | 20.26 | 29.26 | 29.26 | 3.08 | 18.41 |
| Avg. | 13.88 | 16.67 | 4.87 | **30.86** | **24.01** | **24.54** | **5.13** | 33.92 |

# D Extended InnovatorBench Examples

> ### Example of task 14's description
>
> ## Motivation
>
> Reinforcement Learning (RL) training for Large Language Models often suffers from ∗∗entropy collapse∗∗, where the model's output distribution becomes overly deterministic early in training. This severely limits exploration and prevents the model from discovering diverse reasoning paths. Understanding and mitigating entropy collapse is crucial for successful long−form reasoning tasks where exploration of different solution strategies is essential.
>
> ## Task
>
> ∗∗Your task is to implement a new strategy for GRPO in language model reinforcement learning in order to get the highest accuracy and prevent entropy collapse.∗∗

We provide a GRPO algorithm for you as background knowledge. For a specific question–answer pair $(q, a)$, the behavior policy $\pi_\theta^{\mathit{old}}$ samples a group of $G$ individual responses $\{o_i\}_{i=1}^G$. Then, the advantage of the i–th response is calculated by normalizing the group–level rewards $\{R_i\}_{i=1}^G$:

$$
\nabla_\theta J_{GRPO}(\theta) = \mathbb{E}_{(q, a) \sim D, \{o_i\}_{i=1}^G \sim \pi_{\theta_{old}}(\cdot|q)} \left[ \dfrac{1}{G} \sum_{i=1}^G \dfrac{1}{|\mathcal{o}_i|} \sum_{t = 1}^{|\mathcal{o}_i|} \left( \min \left( r_{i,t}(\theta)\hat{A}_{i, t}, \text{clip}(r_{i,t}(\theta), 1 - \epsilon, 1 + \epsilon)\hat{A}_{i, t} \right) \right) \right]
$$ , where $ r_{i,t} = \dfrac{\pi_\theta(o_{i, t}|q,o_{i, <t})}{\pi_\theta^{\mathit{old}}(o_{i, t}|q, o_{i, <t})}$ is the importance sampling ratio.

**Now, you need to implement a variant of GRPO to get the highest accuracy and prevent entropy collapse.**

You should work under the '/workspace/task' and '/workspace/data' directories.

You need to use the original GRPO rollout. Only modify the advantage/loss compute part in '/workspace/task/repositories/verl/verl/trainer/ppo/core_algos.py', do not modify other files in verl.

After you implement the algorithm in '/workspace/task/repositories/verl', you can use '/workspace/task/scripts/train.sh' to train the model based on your code; it will also show the dev set results in the training log. (You can train multiple times, just select the best one to evaluate.)

– Do not modify the parameter in '/workspace/task/scripts/train.sh' except test_freq

Finally, you need to select the best checkpoints in your training, using '/workspace/task/scripts/inference.sh <model_path>' to generate '/workspace/data/outputs/math_results_logits.pt' and '/workspace/data/outputs/math_results.parquet'

### Output Format Requirements:

**File Location**: '/workspace/data/outputs/math_results.parquet'

**File Format**: Parquet format containing model responses

**Required Fields**: The parquet file should contain:
– '"sample_id"' (int): Unique identifier for each test sample (must follow the order in the test set)
– '"output"' (string): The model's complete response, including reasoning and final answer

**Required Output Format for each response:**
```
<reasoning process>
Answer:\boxed{Your Answer}
```

**Logits File**: '/workspace/data/outputs/math_results_logits.pt'
– PyTorch tensor file containing logits for entropy calculation
– Should be a list of tensors, one for each test sample
– Each tensor should have shape '[sequence_length, vocab_size]' where:
   – 'sequence_length': Number of generated tokens for that sample
   – 'vocab_size': Size of the model's vocabulary

**IMPORTANT Notes:**
– Use '/workspace/task/scripts/inference.py' for inference – **do not modify this script**

– The reasoning process should demonstrate clear mathematical thinking
– Final answer must be enclosed in '\boxed{}' format
– Your task is to implement a new strategy for GRPO to prevent entropy collapse and achieve the highest accuracy

The evaluation will check these two files to give the final score based on both accuracy and entropy metrics.

## Data

– **Train set**: Stored in '/workspace/data/datasets/math.8k/train.parquet'
  – data_source: 'math8k'
  – prompt: the math question
  – reward_model: the ground truth answer of the math question

– **Dev set**: Stored in '/workspace/data/datasets/math.8k/dev.parquet'
  – data_source: 'GSM8k' & 'AMC23'
  – prompt: the math question
  – reward_model: the ground truth answer of the math question

– **Test set**: Stored in '/workspace/data/datasets/math.8k/test.parquet'

– **Checkpoint**: Stored in '/workspace/data/checkpoints/Qwen2.5−1.5B'.

## Constraint

– Training time constraint: 24 hours.
– Evaluation time constraint: 10 minutes.
– You can use 8 80GB GPUs.
– You only have 3 chances to evaluate your submission.
– Total time limit: 48 hours.

## Evaluation

**Evaluation metrics:**

1. Accuracy: Measure the accuracy of the variant policy on the test set at the end of the training.
2. Entropy Analysis: Track entropy of the variant policy along the training process.

**Success criteria:**

1. The accuracy of the variant policy on MATH500 should be at least higher than the baseline policy. Try your best to get the highest accuracy!
2. The average entropy of the variant policy should be in a reasonable range.
3. The final score will be the accuracy score * entropy score * 100

## Environment

We have setup the conda environment for you named '/workspace/conda', and we have activated the environment.

## Scripts
You can generate scripts in the '/workspace/task/scripts' directory. You **should not modify scripts** that are originally in the '/workspace/task/scripts' directory.

The following scripts are provided to you; do not modify them:

– '/workspace/task/repositories/verl/scripts/model_merger.py': Given a model path of verl checkpoint, which is a directory containing multiple 'model_world_size_8_rank_{rank_number}.pt' files, you can use this script to merge the model weights into HuggingFace format.
  – Input:
    – '−−local_dir': The path of the verl checkpoint.
  – Output:
    – The checkpoint in HuggingFace format.

## E  LIMITATIONS AND FUTURE WORKS

Despite the advancements brought by InnovatorBench and ResearchGym, there are several areas for improvement in future work:

**Task Diversity** InnovatorBench currently covers a limited set of research tasks. Future work could expand the benchmark to include more diverse, interdisciplinary challenges that reflect real-world scientific research.

**Generalization of Agents** AI agents still show performance variation depending on the model. Further research is needed to improve their generalization across different research tasks and improve transfer learning for broader applicability (Fu et al., 2025; 2024a;b).

**Human-AI Collaboration** The current framework largely focuses on autonomous AI agents. Future work could explore hybrid human-AI workflows, incorporating real-time feedback and collaboration for more realistic research (Ye et al., 2025a).

## F  SUPPORTED ACTIONS OF RESEARCHGYM

We referred to the design of OpenHands (Wang et al., 2024a) and adapted it to the multi-machine, multi-GPU, asynchronous, and other environments required by ResearchGym.

In our settings, the agent is not allowed to run *nohup* or *kill* commands, since we found the agent would like to kill the script to support the multi-machine control (because it seems like a "unrelated" script).

### F.1  COMMAND ACTIONS

The command actions manage terminal session lifecycle and interaction, including session creation, listing, command execution, input/output handling, status inspection, and session termination. The following functions provide comprehensive capabilities to control and operate remote or local computing sessions.

```python
def create_session_action(computer_ip: str = 'localhost', session_id: str
    = None, http_port: int = None, use_proxy: bool = True) -> Dict[str,
    Any]:
    """Create a new terminal session on the computer specified by `
    computer_ip`.

    This function initializes connectivity via `http_port` and `use_proxy
    `. Use `use_proxy=False` for `cpu`/`localhost_cpu` machines and `
    use_proxy=True` for `gpu` machines.

    Args:
```

```
        computer_ip[str]: The IP address of the computer. Default is '
    localhost'.
        session_id[str]: Unique identifier of the target session. If
    absent, a new
        session is created and a new `session_id` is assigned on the host
        `computer_ip`. Default is None.
        http_port[int]: The HTTP port to use to connect to the session.
        use_proxy[bool]: Whether to use a proxy for connecting to the
    session. Set
        `use_proxy=False` for `cpu` and `localhost_cpu` computers, and
    set
        `use_proxy=True` for `gpu` computers. Must align with your
    network topology
        or the connection will fail. Default is True.

    Returns:
        Dict[str, Any]: Dictionary containing session creation status and
        information.
    """

def list_sessions_action(computer_ip: str = None) -> Dict[str, Any]:
    """List all existing sessions.

    Key '<computer_ip>:<session_id>' on the output refers to the session
    <session_id> on <computer_ip>.

    Args:
        computer_ip[str]: The IP address of the computer. If None, lists
    sessions
        on all machines. Default is None.

    Returns:
        Dict[str,
    Any]: Dictionary containing information about all active
        sessions.
    """

def run_command_action(
        command: str,
        computer_ip: str = 'localhost',
        session_id: str = None,
        http_port: int = None,
        wait_for_completion: bool = False,
        use_proxy: bool = True
) -> Dict[str, Any]:
    """Execute a single bash command in the session identified by `
    session_id`.

    If the session does not exist, it will be created and bound to the
    target host (determined by `computer_ip`)
    and will be connected via `http_port` and `use_proxy`. Only one
    command may run concurrently per session.

    Args:
        command[str]: Shell (bash) command to execute in the target
    session's
        working directory and environment.
        computer_ip[str]: The IP address of the computer. Default is '
    localhost'.
```

```
        session_id[str]: Unique identifier of the target session. If
    absent, a new
        session is created on the host determined by `computer_ip`.
    Default is
        None.
        http_port[int]: The HTTP port to use to connect to the session.
    Default is
        None.
        wait_for_completion[bool]: Whether to block until the command
    finishes:
        - True: block up to 10 seconds; on timeout the command process is
     killed.
        - False: return immediately and let the command run in the
    background.
        use_proxy[bool]: Whether to use a proxy for connecting to the
    session. Set
        `use_proxy=False` for `cpu` and `localhost_cpu` computers, and
    set
        `use_proxy=True` for `gpu` computers. Default is True.

    Returns:
        Dict[str,
    Any]: Dictionary containing command execution results and status.
    """

def input_in_session_action(computer_ip: str = 'localhost', session_id:
    str = None, input_text: str = '') -> Dict[str, Any]:
    """Navigate to a webpage based on URL and display its content.

    The environment will cache the webpage content for another action to
    use until perform next web_browse action.

    Args:
        url[str]: The URL to navigate to.
        line_number[int]: The line number to start viewing from. The
    environment
        will perform line_number to line_number+100 lines of content.
    Default is 1.

    Returns:
        Dict[str,
    Any]: Dictionary containing page content and status information.
    """

def get_session_output_action(
        computer_ip: str = 'localhost',
        session_id: str = None,
        start_lines: int = 50,
        end_lines: int = None,
        since_timestamp: float = None
    ) -> Dict[str, Any]:
    """Retrieve the output buffer of the terminal session identified by `
    session_id`.

    If `since_timestamp` is provided, incremental output since that time
    is returned; otherwise, output is sliced by line window (`start_lines
    ` required, `end_lines` optional).

    Args:
```

```
        computer_ip[str]: The IP address of the computer. Default is '
    localhost'.
        session_id[str]: Unique identifier of the target session. The
    session must
        exist and be active. Default is None.
        start_lines[int]: Start offset counted from the end of output
    (>=2).
        Effective only when `since_timestamp` is not set. Usage:
        - `start_lines=N` only: returns the last N lines.
        - With end_lines: returns the slice between `start_lines` and `
    end_lines`.
        end_lines[int]: End offset counted from the end of output (>=1).
    If not
        specified, this tool will return content from the `start_lines`
    to the end
        of the output. If specified, the slice is [start_lines, end_lines
    ):
        inclusive of `start_lines`, exclusive of `end_lines`. Default is
    None.
        since_timestamp[float]: Optional. Fetch output since this Unix
    epoch
        timestamp (seconds, float). When set, it overrides `start_lines`
    and
        `end_lines`. Default is None.

    Returns:
        Dict[str, Any]: Dictionary containing session output and status
        information.
    """

def session_status_action(computer_ip: str = 'localhost', session_id: str
    = None) -> Dict[str, Any]:
    """Get the status of a specific terminal session.

    Args:
        computer_ip[str]: The IP address of the computer. Default is '
    localhost'.
        session_id[str]: Unique identifier of the target session. If
    absent, the
        status of the default session is returned. Default is None.

    Returns:
        Dict[str, Any]: Dictionary containing session status information.
    """

def session_idle_action(computer_ip: str = 'localhost', session_id: str =
    None) -> Dict[str, Any]:
    """Check if a specific terminal session is idle.

    Args:
        computer_ip[str]: The IP address of the computer. Default is '
    localhost'.
        session_id[str]: The ID of the session to check whether it is
    running some
        command or whether it is idle. Default is None.

    Returns:
        Dict[str,
    Any]: Dictionary containing session idle status information.
```

```python
    """

    def clear_session_buffer_action(computer_ip: str = 'localhost', session_id
        : str = None) -> Dict[str, Any]:
        """Clear the output buffer of a specific terminal session.

        The output buffer is a queue of output lines, it will automatically
        clean if the total lines exceed 10000 lines, regardless of using this
         action or not.

        Args:
            computer_ip[str]: The IP address of the computer. Default is '
        localhost'.
            session_id[str]: The ID of the session to clear the output buffer.

        Returns:
            Dict[str,
        Any]: Dictionary containing operation status information.
        """

    def close_session_action(computer_ip: str, session_id: str) -> Dict[str,
        Any]:
        """Close a specific terminal session and kill all sub-processes in
        the session.

        Args:
            computer_ip[str]: The IP address of the computer.
            session_id[str]: The ID of the session to close.

        Returns:
            Dict[str,
        Any]: Dictionary containing operation status information.
        """

    def close_all_sessions_action(computer_ip: str = None) -> Dict[str, Any]:
        """Close all sessions on a specific machine or all machines.

        If you want to close all sessions on a specific machine, you should
        set the `computer_ip`.

        Args:
            computer_ip[str]: The IP address of the computer. If None, closes
        sessions
            on all machines. Default is None.

        Returns:
            Dict[str,
        Any]: Dictionary containing operation status information.
        """

    def kill_session_processes_action(computer_ip: str = 'localhost',
        session_id: str = None, force: bool = False) -> Dict[str, Any]:
        """Kill all processes on a specific session.

        Args:
            computer_ip[str]: The IP address of the computer. Default is '
        localhost'.
            session_id[str]: The ID of the session to kill all processes.
            force[bool]: Whether to force to kill all processes. Default is
        False.
```

```
Returns:
    Dict[str,
Any]: Dictionary containing operation status information.
"""
```

## F.2 BROWSE ACTIONS

The browse actions enable webpage navigation, viewing, scrolling, in-page keyword search, iterative result traversal, and hyperlink extraction from cached web content. Specifically, web_page_goto_action, web_page_goto_line_action, web_page_scroll_down_action, web_page_scroll_up_action, web_page_search_action, web_page_search_next_action, and web_page_get_links_action collectively provide a unified interface for interacting with and extracting information from web pages.

```python
def web_page_goto_action(url: str, line_number: int = 1) -> Dict[str, Any
    ]:
    """Navigate to a webpage based on the given URL and display its
    content.

    The environment will cache the webpage content for subsequent actions
     until another web browsing action is performed.

    Args:
        url[str]: The URL to navigate to.
        line_number[int]: The line number to start viewing from (1-indexed
    ).
        The environment will provide content from line_number to
    line_number+100.

    Returns:
        Dict[str,
    Any]: Dictionary containing page content and status information.
    """

def web_page_goto_line_action(line_number: int) -> Dict[str, Any]:
    """Jump directly to a specific line in the currently cached webpage.

    Args:
        line_number[int]: The line number to jump to (1-indexed).

    Returns:
        Dict[str,
    Any]: Dictionary containing page content and status information.
    """

def web_page_scroll_down_action() -> Dict[str, Any]:
    """Scroll down the currently cached webpage by a fixed number of
    lines.

    This displays the subsequent 100 lines of content.

    Returns:
        Dict[str,
    Any]: Dictionary containing page content and status information.
    """

def web_page_scroll_up_action() -> Dict[str, Any]:
    """Scroll up the currently cached webpage by a fixed number of lines.

    This displays the previous 100 lines of content.
```

```python
    Returns:
        Dict[str,
    Any]: Dictionary containing page content and status information.
    """

def web_page_search_action(keyword: str, context_lines: int = 5) -> Dict[
    str, Any]:
    """Search for a keyword in the currently cached webpage and return
    surrounding context.

    The search returns the first occurrence of the keyword along with the
     specified
    number of context lines.

    Args:
        keyword[str]: The keyword to search for.
        context_lines[int]: Number of context lines to display around each
     match.

    Returns:
        Dict[str, Any]: Dictionary containing search results and status
        information.
    """

def web_page_search_next_action(context_lines: int = 5, search_index: int =
     None) -> Dict[str, Any]:
    """Advance to the next (or specified) search result in the cached
    webpage.

    If search_index exceeds the number of matches, it wraps using modulo
    arithmetic.

    Args:
        context_lines[int]: Number of context lines to display around the
    match.
        search_index[int]: Index of the search result to jump to. If None,
     advances
        to the next result.

    Returns:
        Dict[str, Any]: Dictionary containing search results and status
        information.
    """

def web_page_get_links_action(page_size: int = 10, page_number: int = 1) ->
     Dict[str, Any]:
    """Extract hyperlinks from the currently cached webpage.

    Args:
        page_size[int]: Number of links to return per page. Default is 10.
        page_number[int]: The page number of results to display. Default
    is 1.

    Returns:
        Dict[str,
    Any]: Dictionary containing link list and status information.
    """
```

## F.3    FILES ACTIONS

The file manipulation module provides capabilities to navigate, inspect, create, modify, and search files or directories. It includes editing **file_edit_action**, opening and navigating within files **open_file_action**, **goto_line_action**, **file_scroll_down_action**, **file_scroll_up_action**, creating new files **create_file_action**, searching directories or files **search_dir_action**, **search_file_action**, **find_file_action**, listing directory contents **list_files_action**, and retrieving metadata about the current file **get_file_info_action**. Together these operations provide a complete toolkit for programmatic file system interaction.

```python
def file_edit_action(path: str, start_line: int, end_line: int, content:
    str) -> Dict[str, Any]:
    """Edit a file given path.

    The file's [start,end] lines will be edited to the content. Remember
    this edit
    will change the file's line-linenumber index, so do not edit
    consecutively
    until you use `read_file` tools to read the new file version.

    Args:
        path[str]: The path to the file to edit.
        start_line[int]: The starting line to be edited (including).
        end_line[int]: The ending line to be edited (including).
        content[str]: The content to be written or edited in the file. It
     will
        replace the content between `start` and `end` lines.

    Returns:
        Dict[str, Any]: Dictionary containing edit operation status and
        information.
    """

def open_file_action(path: str, line_number: int = 1, context_lines:
    Optional[int] = None) -> Dict[str, Any]:
    """Open a file and display its content around a specific line.

    The environment will cache the file content for another file action
    to use until perform next open_file action.

    Args:
        path[str]: The path to the file to open.
        line_number[int]: The line number to focus on (1-indexed). Default
     is 1.
        context_lines[Optional[int]]: Number of lines to show as context.
    Default is
        None (uses default window size).

    Returns:
        Dict[str,
    Any]: Dictionary containing file content and status information.
    """

def goto_line_action(line_number: int) -> Dict[str, Any]:
    """Jump to a specific line in the currently open file and show the
    content around the line.

    Args:
        line_number[int]: The line number to jump to (1-indexed).
```

```
    Returns:
        Dict[str,
    Any]: Dictionary containing file content and status information.
    """

def file_scroll_down_action() -> Dict[str, Any]:
    """Scroll down 100 lines in the currently open file.

    Returns:
        Dict[str,
    Any]: Dictionary containing file content and status information.
    """

def file_scroll_up_action() -> Dict[str, Any]:
    """Scroll up 100 lines in the currently open file.

    Returns:
        Dict[str,
    Any]: Dictionary containing file content and status information.
    """

def create_file_action(filename: str, content: str = "") -> Dict[str, Any
    ]:
    """Create a new file with the specified content.

    It will also replace the original file if it already exists.

    Args:
        filename[str]: The name/path of the file to create.
        content[str]: The content to write to the new file. Default is
    empty string.

    Returns:
        Dict[str,
    Any]: Dictionary containing file creation status and information.
    """

def search_dir_action(search_term: str, dir_path: str = './') -> Dict[str,
     Any]:
    """Search for a text pattern in all files within a directory.

    Args:
        search_term[str]: The text to search for.
        dir_path[str]: The directory path to search in. Default is current
     directory.

    Returns:
        Dict[str, Any]: Dictionary containing search results and status
        information.
    """

def search_file_action(search_term: str, file_path: Optional[str] = None)
    -> Dict[str, Any]:
    """Searches for a text pattern in a specific file or the currently
    open file.

    Args:
        search_term[str]: The text to search for.
        file_path[Optional[str]]: The file path to search in. If None,
    searches in
```

```
        currently open file. Default is None.

    Returns:
        Dict[str, Any]: Dictionary containing search results and status
        information.
    """

def find_file_action(file_name: str, dir_path: str = './') -> Dict[str,
    Any]:
    """Finds files by name pattern within a directory.

    Args:
        file_name[str]: The file name or pattern to search for.
        dir_path[str]: The directory path to search in. Default is current
    directory.

    Returns:
        Dict[str, Any]: Dictionary containing search results and status
        information.
    """

def list_files_action(path: str = ".", show_hidden: bool = False) -> Dict[
    str, Any]:
    """List all files and directories in a specified path.

    Args:
        path[str]: The directory path to list contents of. Default is
    current directory.
        show_hidden[bool]: Whether to show hidden files/directories.
    Default is
        False.

    Returns:
        Dict[str,
    Any]: Dictionary containing directory listing and status
        information.
    """

def get_file_info_action() -> Dict[str, Any]:
    """Get information about the currently open file.

    Returns:
        Dict[str,
    Any]: Dictionary containing file information and status.
    """
```

### F.4 SEARCH ACTIONS

The search functionality provides web-based information retrieval capabilities: search_action issues queries to external search engines (e.g., Google or Bing) and returns up to top_k ranked results along with associated status metadata; result sets are capped to prevent excessive retrieval.

```
def search_action(query: str, top_k: int = 10) -> Dict[str, Any]:
    """Perform a web search using engines such as Google or Bing.

    Args:
        query[str]: The search query to look up on the web.
        top_k[int]: The maximum number of search results to return.
        If the number exceeds 100, it will be set to 100. Default is 10.
```

```
Returns:
    Dict[str, Any]: Dictionary containing search results and status
    information.
"""
```

## F.5 PARSER ACTIONS

This set of parser actions collectively enables the extraction and transformation of information from diverse input modalities. Specifically, **parse_pdf_action**, **parse_docx_action**, **parse_latex_action**, and **parse_pptx_action** handle the parsing of structured document formats, while **parse_audio_action**, **parse_image_action**, and **parse_video_action** process unstructured multimedia inputs such as speech, images, and video, thereby supporting a unified mechanism for multimodal content understanding and storage.

```
def parse_pdf_action(file_path: str, save_path: str) -> Dict[str, Any]:
    """Parse a PDF file, extract text content and save to a file.

    Args:
        file_path[str]: The path to the PDF file to parse.
        save_path[str]: The path to save the parsed content.

    Returns:
        Dict[str,
    Any]: Dictionary containing parsing status and information.
    """

def parse_docx_action(file_path: str, save_path: str) -> Dict[str, Any]:
    """Parse a DOCX file and save the parsed content to a file.

    Args:
        file_path[str]: The path to the DOCX file to parse.
        save_path[str]: The path to save the parsed content.

    Returns:
        Dict[str,
    Any]: Dictionary containing parsing status and information.
    """

def parse_latex_action(file_path: str, save_path: str) -> Dict[str, Any]:
    """Parse a LaTeX file and save the parsed content to a file.

    Args:
        file_path[str]: The path to the LaTeX file to parse.
        save_path[str]: The path to save the parsed content.

    Returns:
        Dict[str,
    Any]: Dictionary containing parsing status and information.
    """

def parse_audio_action(file_path: str, save_path: str, model: str = '
    whisper-1') -> Dict[str, Any]:
    """Parse an audio file, transcribe its content and save the parsed
    content to a file.

    Args:
        file_path[str]: The path to the audio file to parse.
        save_path[str]: The path to save the parsed content.
        model[str]: The model to use for audio transcription.
```

```
    Returns:
        Dict[str,
    Any]: Dictionary containing parsing status and information.
    """

def parse_image_action(file_path: str, save_path: str, task: str = '
    Describe this image.') -> Dict[str, Any]:
    """Parse an image file, analyze its content and save the parsed
    content to a file.

    Args:
        file_path[str]: The path to the image file to parse.
        save_path[str]: The path to save the parsed content.
        task[str]: The task description for image analysis.

    Returns:
        Dict[str,
    Any]: Dictionary containing parsing status and information.
    """

def parse_video_action(file_path: str, save_path: str, task: str = '
    Describe this image.', frame_interval: int = 30) -> Dict[str, Any]:
    """Parse a video file, analyze its content and save the parsed
    content to a file.

    Args:
        file_path[str]: The path to the video file to parse.
        save_path[str]: The path to save the parsed content.
        task[str]: The task description for video analysis.
        frame_interval[int]: The frame interval for video analysis.
    Default is 30.

    Returns:
        Dict[str,
    Any]: Dictionary containing parsing status and information.
    """

def parse_pptx_action(file_path: str, save_path: str) -> Dict[str, Any]:
    """Parse a PPTX file and extract text content.

    Args:
        file_path[str]: The path to the PPTX file to parse.
        save_path[str]: The path to save the parsed content.

    Returns:
        Dict[str,
    Any]: Dictionary containing parsing status and information.
    """
```

## F.6 SPECIAL ACTIONS

The special actions include null_action for performing no operation, think_action for recording the agent's thoughts, eval_action for submit the result and gain the score, view_hint_action for inspecting task-related hints with an associated score penalty, and finish_action for terminating the research task.

```
def null_action() -> str:
    """Null Action.

    Returns:
```

```
        "No Action"
    """

def think_action(action: BaseAction) -> BaseObservation:
    """Handle an action where the agent logs a thought.

    This function processes the ThinkAction and returns the thought as an
     observation.

    Args:
        action[BaseAction]: The ThinkAction to handle.

    Returns:
        BaseObservation: Observation containing the thought and status
        information.
    """

def view_hint_action(action: BaseAction) -> BaseObservation:
    """View the hint for the current task.

    Some tasks contain hints, this function allows the agent to view the
    hint,
    but using this action will deduct the agent's score.

    Args:
        action[BaseAction]: The ViewHintAction to handle.

    Returns:
        BaseObservation: Observation containing the hint content and
    status
        information.
    """

def eval_action() -> None:
    """
    An action where the agent evaluates the agent's output (some files
    and the content inside the files), which is declared in the task
    description  (original task instead of subgoal). The argument of this
     action should be empty, do not add any key inside the argument
    """

def finish_action() -> None:
    """
    Terminating the research task.
    """
```

# G PROMPT USED IN AGENTS

## G.1 SUMMARY

**System prompt for summarizing the internal research history**

You are the component that summarizes the internal research history into a given structure for an AI Innovator agent.

When the research history grows too large, you will be invoked to distill it into a concise, structured XML snapshot. This snapshot is CRITICAL, as it will become the agent's *only* memory of the

past. The agent will resume its research based solely on this snapshot. All crucial details, hypotheses, experimental plans, results, learnings, and user directives MUST be preserved.

First, you should think through the entire history in a private <history>. Review the overall research goal, the agent's experiments, code modifications, tool outputs, and experimental results. Identify every piece of information that is essential for future research steps.

After your reasoning is complete, generate the final <state_snapshot> XML object. Be incredibly dense with information. Omit any irrelevant conversational filler.

# Context Overview

You will be given the following contexts:
1. The original task description, which is at the beginning of the context.
2. The history, it may contains 2 parts:
    2.1 Your reaction towards the observation from the environment, and its corresponding observation from the environment.
    2.2 Your summary of some parts of the action–observation history. (Since the action–observation history is too long, you just summarize some parts of it.)

# Input Context Format

For easier understanding, the user will place the key factors in the following format:

1. The original task description:
<task_description>
YOUR TASK DESCRIPTION
</task_description>

2. The history you need to summarize:
<history>
...
</history>

## Real User

– If context is provided in the <real_user></real_user> tag, you should perform reflection and save your reflection results in <reflection></reflection> (at least n reflections for n <real_user> entries).
– The real user's advice must be treated as IMPORTANT.

The structure of your output is specified in 'internal_summarize' tool, you MUST follow the tool's instruction.

Try your best to make this summary!

---

## Tool prompt for summarizing the internal research history

# The structure of 'summary_content' MUST be as follows:

```
<state_snapshot>
    <state_of_the_art>
        <!-- The SOTA benchmark to surpass. -->
        <!-- Example: "The current SOTA score is 0.85. We need to beat this." -->
    </state_of_the_art>
```

```
<hypotheses>
    <!-- List of active, tested, or pending hypotheses. -->
    <!-- Example:
    - [TESTING] Hypothesis 1: Adding a penalty for verbosity in the reward function will
    improve conciseness without harming helpfulness.
    - [PROVEN] Hypothesis 2: Normalizing rewards by batch statistics stabilizes training.
    - [TODO] Hypothesis 3: Using data augmentation on the prompt dataset will increase
    instruction-following capabilities.
    -->
</hypotheses>

<key_knowledge>
    <!-- Crucial facts, takeaways, and constraints the agent must remember based on the
    conversation history and interaction with the user. Use bullet points. -->
    <!-- Example:
    - Ray: ray has started with \'ray start --head\' but havn't check its status.
    - API Endpoint: The primary API endpoint is \'https://api.example.com/v2\'.
    - Learning rate > 1e-4 causes training instability.
    - The main dataset is located at '/data/datasets/rl_dataset_v2.parquet'.
    - Model weights are at '/data/checkpoints/base_model.pth'.
    - Trainging models: llamafactory-cli has been started, the response of the training data is
    generated by the Qwen2.5-72B-Instruct model.
    - The number of remaining calls to the 'eval' tool is 2.
    - Reading File: The 'test.parquet' data's value is too long, I should read the special key
    -->
</key_knowledge>

<reflection>
    <!-- Reflection that the agent should remember based on conversation history and
    interactions. Use bullet points. -->
    <!-- Present the reasoning step concisely when stating an Reflection. -->
    <!-- Each line should be in the format of: 'Reflection: concise reasoning step and its
    corresponding facts in the history. -->
    <!-- Only add, edit or merge reflection when there are some incidents in the history. Do
    not generate redundant reflections. -->
    <!-- The reflections should be general. -->
    <!-- Add reflection from below examples when they appear in the history; you are
    encouraged to create new, relevant reflection or edit reflection towards new situation. -->
    <!-- If this reflection comes from real user's advice (content inside <real_user> tag), cite
    its input in [real_user][/real_user]. -->
    <!-- Examples:
    - Use a special key to read file: in 'test.parquet', some values are very long; reading directly
     may exceed the context length.
    - Use 'wait_for_completion=False' for Ray/training/inference jobs lasting >10 seconds; in
    the past, jobs were killed when 'wait_for_completion=True'.
    - Check GPU status before training/inference: once, training started while another process
    was already running, causing confusion and wasted time debugging the conda environment.
    [real_user]Do not running this inference scripts. You have already run another training
    scripts[/real_user]
    - Always check the file after editing to avoid unexpected modification.
    - Run commands in the correct path: if not run in folder 'A', Python may import the
    environment's 'math' module instead of 'A/math.py', even with 'sys.path.append('A')'.
    - Be patient: importing 'transformers' or starting Ray can take about 5 minutes; avoid
    killing the process prematurely. [real_user]Your training script is right, why you kill this
    script?[/real_user]
```

```
        – Do not specify 'end_lines' in most cases: you often need to read the tail of the session to
        get the newest information.
        – Determine scope: only the information after the last exception log or interactive prompt is
        the last command's output; confusion often happens when 'start_lines' is set too large.
        – Check the session's status and kill unused sessions after planning/summarization: a run
        was started and forgotten, leading to duplicate launches; verify idleness and the latest output
        before starting again.
        -->
</reflection>

<file_and_browser_state>
        <!-- List files that have been created, read, modified, deleted and key data artifacts. Note
        their status and critical learnings. -->
        <!-- Example:
        – CWD: '/workspace/task/'
        – MODIFIED: '/workspace/task/reward.py' – Implemented the verbosity penalty.
        – CREATED: '/workspace/task/scripts/data_augmentation.py' – Script to apply back–
        translation.
        – DATASET: '/workspace/data/datasets/augmented_prompts.json' – New dataset created
        from Hypothesis 3.
        – READING: \'README.md\' – The last file you are opening/reading.
        – BROWSED: \'https://www.google.com/search?q=new+feature\' – The last browswe
        page you have visited.
        -->
</file_and_browser_state>

<recent_sessions>
        <!-- List **all** sessions that have been created and not been closed. Note their status and
        critical learnings. -->
        <!-- Only the session maybe running will have GPU usage. If the running is finish, GPU
        usage should be None. -->
        <!-- Idle means there is no process running in this session, if one process is end and not
        run other command in the session, this session is idle -->
        <!-- Highlight the GPU that may have conflict in different session -->
        <!-- Example:
        – [session ID1] Last command: [Command in session ID1], Idle: False, GPU usage:
        computer ip xxx.xxx.xxx.xxx GPU 0,1,2,3,4,5,6,7 and computer ip xxx.xxx.xxx.xxx GPU
        0,1,2,3,4,5,6,7
        – [session ID2] Last command: [Command in session ID2], Idle: True, GPU usage: None
        – [session ID3] Last command: [Command in session ID3], Idle: False, GPU usage:
        computer ip xxx.xxx.xxx.xxx GPU 0,1,2,3
        -->
</recent_sessions>

<recent_actions>
        <!-- A summary of the last few significant agent actions and their outcomes. Focus on
        facts. -->
        <!-- Example:
        – Ran \'grep 'old_function'\' in session xxxxxxxx, computer ip xxx.xxx.xxx.xxx which
        returned 3 results in 2 files.
        – Ran \'bash inference.sh\' in session xxxxxxxx, computer ip xxx.xxx.xxx.xxx, which
        failed due to the incorrect output data path.
        – Ran \'ls –F static/\' in session xxxxxxxx, computer ip xxx.xxx.xxx.xxx and discovered
        image assets are stored as \'.webp\'.
        – Ran \'bash train.sh\' in session xxxxxxxx, computer ip xxx.xxx.xxx.xxx, it is still
        running now.
        -->
```

```
    </recent_actions>

    <experiment_history>
        <!-- A summary of the last few significant experiments and their outcomes. -->
        <!-- Example:
        − Experiment 1 (Hypothesis 1): Ran training with verbosity penalty. Result: Alignment
          score increased to 0.86, but helpfulness dropped slightly. See logs in '/workspace/task/logs/
          exp_1/'.
        − Experiment 2 (Hypothesis 2): Implemented reward normalization. Result: Training was
          stable, loss converged faster. Final score was 0.84. See logs in '/workspace/task/logs/exp_2
          /'.
        -->
    </experiment_history>

</state_snapshot>
```

## G.2 REACT

### ReAct system prompt

You are an interactive AI Innovator. Your primary goal is to autonomously conduct cutting−edge AI research (e.g. designing novel models and algorithms, optimizing training processes, and finding new datasets). The user will provide you a task description and a base codebase to guide your research. Your mission is to code, experiment, and analyze the results to produce innovative solutions, which surpass the current state−of−the−art.

# Core Mandates

− **Scientific Rigor:** Approach every task with a researcher's mindset. Formulate clear hypotheses, design controlled experiments, and draw conclusions based on empirical evidence.
− **Conventions:** Rigorously adhere to existing project conventions when reading or modifying code. Analyze surrounding code, configurations, and documentation first.
− **Plan−First Rule:** For every new task or scope change, create a concise, structured plan before any code edits, training, or long commands. Always decompose the task into smaller subgoals. Use the 'think' tool by default. If the direction is ambiguous or deviates materially from the goal, use the 'think' tool again to refine the plan.
− **Libraries/Frameworks:** NEVER assume a library/framework is available or appropriate. Verify its established usage within the project (check imports, configuration files like 'pyproject.toml', 'requirements.txt', etc.) before employing it. Prioritize using the existing environment to ensure reproducibility.
− **Style & Structure:** Mimic the style (formatting, naming), structure, and architectural patterns of existing code in the project.
− **Idiomatic Changes:** When editing, understand the local context (imports, functions/classes) to ensure your changes integrate naturally and idiomatically. Check the file via 'open_file' after editing.
− **Error Handling:** On exceptions, fail fast and raise immediately; log clear error messages including key variable values, function arguments, and stack traces; handle errors at the appropriate abstraction layer with reproducible debugging context; never silently ignore exceptions or log vague messages like 'Error occurred'; add print function to show the key variable values, function arguments that may realted to the bug.
− **Comments:** Add code comments sparingly. Focus on *why* something is done, especially for complex algorithms or non−obvious logic, rather than *what* is done.
− **Proactiveness & Exploration:** Thoroughly investigate the research problem. This includes exploring the data, trying different hyperparameters, and considering alternative approaches beyond the most obvious path.

– **Confirm Ambiguity/Expansion:** Before undertaking large−scale experiments or significant deviations from the core research goal, THINK TWICE. However, avoid overthinking; actively putting your thought into practice.
– **Explaining Changes:** After completing an experiment or code modification, provide a concise summary of the changes and the key results.
– **Path Construction:** Before using any file system tool, construct the full absolute path for the 'file_path' argument.
– **Do Not Ever Revert Changes:** Do not revert changes unless they cause an error or you are instructed to do so.
– **Do Not Modify the Provided Datasets and Checkpoints:** Do not modify the provided datasets and checkpoints. If you want to change some data, you need to save a backup.
– **Always Try Your Best & Never Give Up:** The user provides you with the state−of−the−art results in task description. TRY YOUR BEST to surpass the state−of−the−art in the research field. Never terminating the task unless you get full mark (100 score) in the evaluation.
– **Be PATIENT:** Use 'check_session_idle' to check if these is subprocess running in a given session and use 'get_session_output' to check the outputs. It may takes **serveral minutes** to load a single package. Do not kill it at first. Notice that sometimes the output returned from 'get_session_output' is not displayed correctly. The subprocess information returned from 'check_session_idle' is usually correct.
– **Seperate the information:** Only the information after the last excpetion log or interactive prompt is the last command's output. Ignore the information before the last excpetion log or interactive prompt if you only want to check the last command's sitiuation.

# Primary Research Workflow

When requested to perform AI research tasks (e.g., design a reward function, augment or clean data, collect new datasets, improve a loss function, build a workflow), follow this sequence:

1. **Understand & Hypothesize:**
    – Deeply analyze the task description, including motivation, task (research goal), the provided codebase (scripts), the provided datasets (if available), resource constraints, and evaluation metrics.
    – Use tools like 'open_file', 'search_file', 'find_file', 'search_dir', 'list_files', 'get_file_info' to explore the codebase, understand file structures, existing code patterns, and conventions.
    – Use shell commands or specialized scripts to inspect the data (e.g., check shape, distribution, examples). However, do not modify the provided datasets. If the data length is too long (e.g., greater than 30000 characters), you should try another way to inspect it (e.g., read the value of some specidied key).
    – Formulate a clear, testable hypothesis. For example: "Hypothesis: Augmenting the SFT data with back−translation will improve model performance on task X." or "Hypothesis: A new loss function incorporating term Y will lead to faster convergence."

2. **Plan & Design Experiment:**
    – Build a coherent and grounded plan (based on the understanding and hypothesis in step 1) for how you intend to resolve the user's task.
    – MUST use the 'think' tool to generate the experimental plan. Do not generate plan by yourself.
    – Specify the exact implementation changes required (e.g., data processing steps, code modifications for the model or training loop).
    – Outline the training procedure (hyperparameters, number of epochs) and the evaluation protocol (metrics, dev set, test set).
    – Consider the remaining working time and the resource constraints to design the experiment.
    – Share an extremely concise yet clear plan with the user if it would help the user understand your thought process.
    – If the historical plan is too high−level or not actionable, call the 'think' tool again to break it down into executable subtasks and milestones.

3. **Implement:**

– Use the available tools (e.g., 'edit_file', 'open_file', 'run_command', 'create_file') to implement the changes.
– Incremental Progress over Big Bangs: Always make minimal edits/additions to the codebase.
– After editing or implementing changes, always check the edits/addionts to make sure they are bug free. You can't use edit_file until you read the place you want to edit. Since once you edit, the line number towards the context will be changed.
– You **MUST** read the place you want to change before you edit the file. You **MUST** check the edit result after executing the 'edit_file' action. You **MUST NOT** doing consecutive edit.
– Write or modify scripts only when user–provided task description requires you to do so. Adhere strictly to the project's established conventions.

4. **Train & Execute:**
– Start ray before verl training (And never kill this process).
– Run the training script using 'run_command'. Be mindful that this may be a long–running process (e.g., training a LLM model). Use background execution if necessary.
– Use 'get_session_output' to check the training output (If you want to get the newest output, do not specify the 'end_lines')
– Check the GPU status (via 'nvidia–smi' and 'ray status') before training, there will be a default 700–4000M VRAM usage for other program. If you find the VRAM usage is bigger than this number, you should list all sessions by using 'list_sessions' and check whether each session is idle or is running some script. If the session is idle and you no longer use it, you should remember the experience you gained from this session and close this session. If the session is busy, you need to choose one of the following actions based on the execution: (1) wait for the training to finish via 'sleep' for most of the time. (2) kill this session if the training time is longer than the '<remaining_working_time>' (3) Do other things (e.g. use other empty GPU to do inference).
– Assign a new training process to a GPU only if its available VRAM is greater than the process's required VRAM; otherwise, do not start the process on that GPU. (In most of the time, if the GPU's VRAM usage is greater than 10000M, this GPU is not available)
– Monitor the logs to ensure the experiment is running as expected and to catch any errors early.
– After training has truly started (logs show "compute loss / backprop"), wait 5–10 steps to stabilize throughput, then estimate the remaining training time ETA from recent average step time. If ETA exceeds the remaining working time, terminate (kill) the training process by 'kill_session_process' tool.
– **Always be patient and do not interfere the normal training process. Do not perform any inference before the training completes.**
– If there are previous checkpoints, you can load it to accelerate the training process.
– Training process may costs several hours to days, be patient.

5. **Analyze & Infer:**
– Use 'get_session_output' to get session output periodically.
– Use 'check_session_idle' to check whether the session is idle. If the session is not idle, additional information of the children processes will be given to you.
– Once training is complete (either when a completion signal is received or the final checkpoint is persisted), immediately use 'run_command' to execute the inference scripts on the dev/test datasets to collect results.
– If the task does not provide inference scripts, generate them yourself.
– Do not run inference while training is still ongoing. It will make the training process unstable (even kill the training)
– Dev datasets are used to evaluate the performance of the model. You can use dev datasets to evaluate the performance of the model by yourself.
– Analyze the output: compare evaluation metrics, examine loss curves, and inspect model outputs.
– Analyze using the given script if one is provided. If no script is provided, save the context as a file and run it when the context exceeds 10 lines.

   – If the data you want to read is in json/jsonl/parquet/pandas format, always read the head/key of the data first, since their value may be very long!

6. **Evaluate:**
   – ** Cherish the opportunity to evaluate.** You only have {PromptBuilder.task_config. max_eval_num} chances to evaluate the results. When all {PromptBuilder.task_config. max_eval_num} chances are used up, you can still work but you do not have any evaluate chance.
   – You MUST run the inference script to generate results on test datasets before submitting the results.
   – The results MUST be saved in the '/workspace/data/outputs' directory.
   – Strictly validate that the format of output data ('/workspace/data/outputs') conforms to the task description.
   – When you are sure that the results on test datasets can be submitted, use the 'eval' tool to submit the results.
   – Backup all your output in output files to other place with its corrposing score after evaluation, and select the best output files when you want to finish your task.

7. **Conclude & Iterate:**
   – Summarize the experiment's findings and results. Did the experimental results surpass the state −of−the−art? Was the hypothesis supported? Why or why not?
   – Present the key results and artifacts (e.g., log files, metric charts) to the user.
   – Based on the outcome, propose the next steps: a refined hypothesis for a new experiment, a suggestion to adopt the new change, or a conclusion that the approach was successful.
   – You MUST save the evaluation result that gets the highest score (maybe surpass SOTA) in '/ workspace/data/outputs' directory.
   – **Always keep fighting until the evaluation score of the output data ('/workspace/data/outputs ') is 100.**

# Operational Guidelines

## Sleep During Long Training and Inference.
– Call 'sleep' for 5−10 minutes when the training just start ($< 1$ step), since it may take a long time to import python packages.
– During the very beginning of training ($< 5$ steps for SFT and $< 2$ steps for RL), allow only short sleeps (less than 120 seconds). After that, take several long sleeps until the training finishes. Do not create any process that uses the same GPU as this training. Do not be afraid of sleeping during training.
– When inference takes several minutes or hours, make sure to call 'sleep'.

## Follow Instructions From Real User
– If context is provided in the <real_user></real_user> tag, follow it.

## Tone and Style
– **Clarity over Brevity (When Needed):** While conciseness is key, prioritize clarity for essential explanations or when seeking necessary clarification if a request is ambiguous.
– **No Chitchat:** Avoid conversational filler, preambles ("Okay, I will now..."), or postambles ("I have finished the changes..."). Get straight to the action or answer.

## Security and Safety Rules
– **Explain Critical Commands:** Before executing commands with 'run_command' that modify the file system, codebase, or system state, you *must* provide a brief explanation of the command's purpose and potential impact. Prioritize user understanding and safety.
– **Security First:** Always apply security best practices. Never introduce code that exposes, logs, or commits secrets, API keys, or other sensitive information.
– **Work under the user's specified working directory:** You should work under the user's specified working directory (e.g., '/workspace'). You should not do anything outside of the working directory.

## Tool Usage
– ∗∗Tools In This Turn:∗∗ Only the tools provided in this turn are available. Do not call, reference, or simulate any tools from earlier turns. They are ∗∗not available∗∗ now.
– ∗∗Think, and then invoke the tool call:∗∗ Before any tool call, you MUST evaluate current sitiuatio , decide which tool is suitable and plan the exact query/inputs.
– ∗∗File Paths:∗∗ Always use absolute paths when referring to files with tools like 'open_file' or 'create_file'. Relative paths are not supported. You must provide an absolute path.
– ∗∗Command Execution:∗∗ Use the 'run_command' tool for running shell commands, such as 'python train.py −−config my_config.yaml' or 'python −c "import pandas as pd; df = pd.read_parquet ('data.parquet'); print(df.head())"'. Remember the safety rule to explain modifying commands first.
– ∗∗Background Processes:∗∗ Use background processes (via \'&\') for commands that are unlikely to stop on their own, e.g. \'node server.js &\'.
– ∗∗Interactive Commands:∗∗ Try to avoid shell commands that are likely to require user interaction (e.g. \'git rebase −i\'). Use non−interactive versions of commands (e.g. \'npm init −y\' instead of \'npm init\') when available, and otherwise you should input the command yourself on the command line on behalf of the user by 'input_in_session' tool.
– ∗∗Being proactive to use tools:∗∗ All tool calls (also denoted as 'function calls' or 'actions') do not require confirmation from the user. You should be proactive to use tools to complete the task.
– ∗∗Output correct format:∗∗ The function will use the default arguments if its argument is not specified. Do not output \"None\" or \"null\" in the output arguments, since their format is string which may disalign with the arguments type.

## Interaction Details
– ∗∗User Instruction:∗∗ When you are in the middle of a task, the user might check the progress of the task and give some feedback. Once you receive the feedback, you should follow the user's instruction to continue to complete the task.

## Environment Information
– ∗∗WORKSPACE:∗∗ Your WORKSPACE is located at '{PromptBuilder.task_config.workspace}'. The WORKSPACE is shared between different computers.

## Computer Configuration
– ∗∗Computer Pool:∗∗ We have provided you with {len(PromptBuilder.task_config.computer_pool)} computers with different types, which are:
{computer_pool_str}
  – 'cpu' computers are remote computers with CPU, 'localhost_cpu' is the local computer with CPU, and 'gpu' computers are remote computers with GPU.
  – You are only premitted to use the GPU in 'gpu' computers, do not use it or running some related command (for example 'ray start') in 'localhost_cpu' or 'cpu' computers.
  – 'gpu' computers can never connect 'localhost_cpu' or 'cpu' computers via internet (for example 'ping')
  – ∗∗Do not use 'gpu' computer to install any package, because it has no internet connection. It also can't connect the cpu via internet.∗∗

# H   THE USE OF LARGE LANGUAGE MODELS

In the process of drafting this paper, we employed large language models (LLMs) as an auxiliary tool to enhance the quality and clarity of our written English. The primary application was to identify and correct grammatical inaccuracies, refine sentence structures, and polish academic expressions, thereby improving the overall readability and professionalism of the manuscript.

Specifically, selected paragraphs or sentences from our initial drafts were input into an LLM (e.g., DeepSeek-v3.1 or a comparable model) with explicit instructions focused solely on language checking and polishing. The prompts were designed to request grammatical corrections, suggestions for

more concise or academically appropriate phrasing, and improvements in logical flow, without altering the core technical content or scientific meaning.

It is crucial to emphasize that the role of the LLM was strictly limited to that of a writing assistant. All substantive intellectual contributions, including the core ideas, theoretical framework, experimental design, data analysis, and result interpretation, remain entirely our own. The final decision to adopt any suggestion provided by the LLM was always subject to our careful review and judgment. We ensured that every change aligned with our intended meaning and adhered to the standards of academic integrity.

This use of LLMs significantly streamlined the writing and revision process, allowing us to focus more effectively on the scientific rigor and conceptual depth of our work.

