# OpenReview forum: "InnovatorBench: Evaluating Agents’ Ability to Conduct Innovative AI Research"
_ICLR.cc/2026/Conference — ICLR 2026 Poster_

### Official Review · Reviewer_kpnh · 2025-10-17

**Soundness:** 2
**Presentation:** 3
**Contribution:** 1
**Rating:** 2
**Confidence:** 4

**Summary:**

This paper aims to benchmark the capabilities of LLM agents in AI research. First, this paper proposes InnovatorBench, which evaluates the LLM agents in data construction, filtering, augmentation, loss design, reward design , and scaffold construction. Second, this paper proposes ResearchGym, a sandbox for LLM agents to operate in the computers, providing several types of action interfaces. Empirical analyses on InnovatorBench are provided to demonstrate the performance of current LLMs and LLM agents.

**Strengths:**

- Comparison on several LLMs are provided.
- The writing is of good quality.

**Weaknesses:**

- What does the task in InnovatorBench look like? This paper only provides an example of Task 14, with other types of tasks unrevealed.
- The number of the tasks in InnovatorBench is too limited, with only 20 tasks in total for 6 types of different tasks. The empirical findings from such a tiny benchmark can be unreliable. This also makes the reusability of the proposed InnovatorBench limited.
- The experiments on InnovatorBench require computing machines with 8x80 GB GPUs, which are too costly for research use.
- Lack of empirical results of open-sourced light-weight LLMs, such as Qwen3-32B.
- Typos: some references are cited in wrong formats.

**Questions:**

I think I would not change my opinion during discussion phase; so there are no questions.

---

> ### Author Response · Authors · 2025-11-17
>
> We thank the reviewer for their insightful comments. We firmly believe that this work offers substantial contributions to the field. Our InnovatorBenchmark is more challenging and realistic, as it evaluates the agent's ability to autonomously generate ideas and iteratively refine them. Furthermore, our ResearchGym extends the capabilities of agents to a larger scale through snapshotting and multi-node control.
>
> Here are our responses to the comments.
>
> **Comment 1:** What does the task in InnovatorBench look like? This paper only provides an example of Task 14, with other types of tasks unrevealed.
>
> **Response to comment 1:** **We have added the description of other tasks in the supplementary material, `task_descriptions` folder.** We will also release the tasks publicly. We did not provide all datasets and model checkpoints because they are too large, and giving a HuggingFace link will avoid the double-blind rule.
>
> **Comment 2:** The number of the tasks in InnovatorBench is too limited, with only 20 tasks in total for 6 types of different tasks. The empirical findings from such a tiny benchmark can be unreliable. This also makes the reusability of the proposed InnovatorBench limited.
>
> **Response to comment 2:** Constructing benchmarks like InnovatorBench needs a lot of time and effort. Specifically, each task in InnovatorBench requires an average of **one week** of effort from **a PhD-level expert** to develop the task description, corresponding starter codebase, and evaluation metrics (Please refer to Appendix B for details).
> Furthermore, the inclusion of 20 tasks is consistent with common practice in research agent benchmark design. For instance, both PaperBench [1] also comprises 20 tasks, demonstrating that this scope is well-established in the field.
>
> **Comment 3:** The experiments on InnovatorBench require computing machines with 8x80 GB GPUs, which are too costly for research use.
>
> **Response to comment 3:** We recognize the computational requirements of InnovatorBench, but **believe such demands reflect an inevitable direction in agent research.** As existing benchmarks with simpler tasks (e.g., GSM8K, AlfWorld) become increasingly saturated, **the community will need to progress toward more complex and realistic task environments**—a transition that inherently involves greater resource investment. At the same time, we expect these requirements to ease as technology advances. **A relevant precedent is SWE-Bench [3]: though initially considered costly when introduced two years ago, it is now widely adopted for agent evaluation as computational accessibility has improved.**
>
> **Comment 4:** Lack of empirical results of open-sourced light-weight LLMs, such as Qwen3-32B.
>
> **Response to comment 4:** Actually, it only gained 0 scores. Hence, we decided not to include these results in the final paper.
>
> **Comment 5:** Typos: some references are cited in the wrong formats.
>
> **Response to comment 5:** We have carefully reviewed the manuscript to correct all typos. To ensure all references are properly formatted, could you please specify which citations require correction? We will address them immediately.
>
> **We sincerely appreciate the reviewer's constructive suggestions and believe that the additional explanations significantly improve the quality of our submission. We hope that this provides sufficient reasons to raise the score.**
>
> [1] Starace, G., Jaffe, O., Sherburn, D., Aung, J., Chan, J. S., Maksin, L., ... & Patwardhan, T. (2025). PaperBench: Evaluating AI's Ability to Replicate AI Research. arXiv preprint arXiv:2504.01848.
>
> [2] Chen, Z., Chen, S., Ning, Y., Zhang, Q., Wang, B., Yu, B., ... & Sun, H. (2024). Scienceagentbench: Toward rigorous assessment of language agents for data-driven scientific discovery. arXiv preprint arXiv:2410.05080.
>
> [3] Jimenez, C. E., Yang, J., Wettig, A., Yao, S., Pei, K., Press, O., & Narasimhan, K. (2023). Swe-bench: Can language models resolve real-world github issues?. arXiv preprint arXiv:2310.06770.

---

> > ### Comment · Reviewer_kpnh · 2025-11-20
> >
> > Thank you for the authors’ responses. However, from my perspective, I still maintain my recommendation of rejection. My main concern remains the extremely limited number of tasks included in InnovatorBench.
> >
> > InnovatorBench provides only 20 tasks across 6 task types, with each type containing merely 2–5 tasks. Such a small-scale benchmark offers very limited coverage and is likely to yield biased and unreliable evaluations. For example, if an LLM agent performs best on only four data-construction tasks in InnovatorBench, can we confidently claim its superiority? I believe the answer is no.
> >
> > The authors compare InnovatorBench with PaperBench and ScienceAgentBench and claim that they “also contain 20 tasks,” but this comparison is not accurate. ScienceAgentBench actually includes 102 tasks in total. PaperBench contains 20 tasks within a single task type, whereas InnovatorBench provides fewer than 5 tasks across all types. The authors also reference SWE-Bench, yet SWE-Bench is impactful not only because of its unique application scenario but also because it includes over 2,000 high-quality tasks, ensuring robust evaluation.
> >
> > Therefore, I strongly encourage the authors to substantially expand the benchmark—ideally to at least 20 tasks per task type, or even 100—rather than offering fewer than five tasks per type. A meaningful benchmark must provide sufficient task volume to support reliable, generalizable conclusions. I would look forward to a more comprehensive version of InnovatorBench in the future.
> >
> > For other concerns:
> >
> > - Comment 3: The authors could address this limitation by incorporating tasks beyond the scope of LLM research. Since the title refers to conducting innovative AI Research rather than merely LLM Research, expanding the task set would make the work more consistent with its stated aims.
> >
> > - Comment 4: I believe the zero scores for the Qwen models should also be reported, as they highlight important limitations of open-source models.
> >
> > - Comment 5: In line 865, the authors cite the DeepSeek paper and the Qwen2.5-VL paper when referring to the Qwen2.5 model. I think this is a typo?

---

> ### Author Response · Authors · 2025-11-17
> **Author's opinions about the cutting-edge papers**
>
> Besides the rebuttal, we would like to present our ideas of what an ICLR paper should be like.
>
> We believe an ICLR paper should be cutting-edge and present what most people haven't thought of now, but is important for the future.
>
> Focus on the benchmark track, the benchmark should focus on the future, for example, using the research agent to do research and testing the agents' ability, like InnovatorBench. It may be difficult for most of the agents/models to gain good performance on the benchmark, but they may have better performance 2 years later, like OSWorld or SWE-Bench's trend. (They are not even saturated now.)
>
> If we want to use AI agent to breaking through the boundaries of human cognition, it do need a lot of resource, but we believe the company like OpenAI, Google DeepMind, Meta, ByteDance, Alibaba, Deepseek have the ability to do this in the future (and actually they really have a lot of money and resources to support this project), so only focus on nowadays resource limitation is not a wise idea. From the authors' real experience, these companies have a lot of intern opportunities, so the resource will not hinder the talented PHD students or researchers from pursuing the cutting-edge research.

---

> ### Author Response · Authors · 2025-11-21
>
> Thanks for the reviewer's comments.
>
> Before addressing the specific comments, we would like to reiterate the broader scope of our paper. **This work is not solely a benchmark contribution; it also introduces ResearchGym**, a new and practical scaffold designed for developing and evaluating research agents. ResearchGym supports **Multi-Computer Control, Asynchronous Command Execution, and Snapshots Saving and Loading**, enabling agents to operate in real research environments that require multi-GPU or even multi-node computation. We believe this scaffold is useful for the community to develop a research agent that needs multiple GPUs. Furthermore, our empirical analysis reflects concrete limitations of existing LLMs and agents, which we believe is insightful.
>
> **Question about task quantity:**
>
> We acknowledge the reviewer’s concern about the limited number of tasks. However, we want to clarify the underlying design logic of InnovatorBench.
>
> Although we divided InnovatorBench into different domains, which serve as topical diversity rather than fundamentally different task paradigms, we consider all tasks to belong to **one core task type: evaluating an agent’s ability to conduct long-horizon research by autonomously generating innovative ideas and writing code**. Similarly, although Paperbench has tasks in different domains like RL/DPO, etc., it only has a single task type: reproduce the results in the paper. **Consequently, the design logic of InnovatorBench is aligned with the reviewer’s own interpretation of PaperBench.**  This design philosophy is also consistent with recent agent benchmarks such as Vending-Bench [4], which evaluate models using **one** unified long-horizon task (maximize profits and your bank account balance over the course of one year) without decomposing it into multiple independent tasks, and one of its versions **has been tested in [Gemini 3 Pro](https://deepmind.google/models/gemini/).**
>
> The reason why InnovatorBench and PaperBench only have 20 tasks is that the annotation process is really hard and can't be automated nowadays. As a result, for genuinely challenging research tasks, we believe that **20 carefully designed, high-effort tasks are sufficient to reveal the creativity and robustness of research agents**.
>
> That said, we agree with the reviewer that the benchmark can and should evolve. Our long-term vision is for InnovatorBench to function as a **community-driven platform**, and we are already exploring mechanisms to support external task contributions.
>
> **Response to comment 3:** Thanks for the reviewer's suggestion. We have updated our paper title to "InnovatorBench: Evaluating Agents' Ability to Conduct Innovative LLM Research."
>
> **Response to comment 4:** Thanks for the reviewer's suggestion. We have added such information in Appendix C.1.
>
> **Response to comment 5:** We have fixed this typo in the paper.
>
> [4] Backlund, A., & Petersson, L. (2025). Vending-bench: A benchmark for long-term coherence of autonomous agents. *arXiv preprint arXiv:2502.1584*

---

> ### Comment · Reviewer_kpnh · 2025-11-21
>
> Thanks for the responses. I think the “AI Research” in the title was the most misleading part for me. After revising it to “LLM Research”, the contribution of this paper is much clearer. However, I still have concerns regarding the open-source LLMs. Please provide the precise results (including zeros) for the Qwen models, along with the specific model sizes, instead of only stating that “We do not report other models like Qwen2.5 (Qwen et al., 2025) because they gain 0 points in each task.” In addition, I believe the results for the Qwen models should be included in the main text to more clearly highlight the current limitations of open-source LLMs.
>
> - Minor one: reference in Line 297 seems also wrong.
>
> For the limited number of tasks, I choose to maintain my opinion. If the authors can promise to resolve my concern on open-source LLMs (it would be better if the authors can report results using Qwen3-32B thinking mode) during rebuttal or in the next version, I would increase my score to 6.

---

> > ### Author Response · Authors · 2025-11-22
> >
> > We have fixed that typo and added Qwen3-32b experiment in the paper.
> >
> > We find that Qwen3-32b gains 0, because its context window is too small (40960) and often omits important information when summarizing.

---

> > > ### Comment · Reviewer_kpnh · 2025-11-22
> > >
> > > Thanks for the update. I have increased my score to 6 as a borderline accept.

---

> > > > ### Author Response · Authors · 2025-11-22
> > > >
> > > > Thank you very much for your feedback and for engaging with our work throughout the rebuttal. We sincerely appreciate your constructive suggestions and your willingness to reconsider your evaluation.

---

### Official Review · Reviewer_Hyqf · 2025-10-31

**Soundness:** 4
**Presentation:** 3
**Contribution:** 4
**Rating:** 8
**Confidence:** 5

**Summary:**

The authors present a new benchmark that tests research ability on more complex tasks, which is backed by a new evaluation infrastructure harness that allows for more complex tasks. The work includes evaluations and failure analysis of leading frontier models with a simple agent.

**Strengths:**

* The tasks provided are difficult and realistic, which complements or advances existing benchmarks.
* Research gym genuinely seems very useful to move to a more realistic action space (for e.g. with snapshotting and multi-node control). This allows evaluations to test tasks that are substantially more complex.
* The failure analysis was informative and interesting (notably, I wonder if non-Claude agents performed worse as they were only trained to use specific tools like bash or python).

**Weaknesses:**

* While the results are presented reasonably well, it could be clearer. For example, the fact that the tasks were validated by annotators reproducing the reference score in the reasoning gym version was only present in the Appendix but it was a key question about how valid/reviewed are the tasks.
* It might have been useful to test other scaffolds to ensure that the agents were not substantially worse than they could have otherwise been (e.g. openhands or codex).
- One concern is that all performance on this benchmark will be driven by memorization, since all the tasks are based on existing papers that are in training data. It would be more convincing if the authors presented evidence that this was not a concern (i.e. by looking at successes).

**Questions:**

* are agents allowed to use the snapshot functionality via tool calls?
* will you release the tasks publicly?

nits:
- missing parenthesis line 235.
- missing word line 265

---

> ### Author Response · Authors · 2025-11-17
>
> Thanks for the reviewer's comments. Here are our responses to the comments.
>
> **Comment 1:**  While the results are presented reasonably well, it could be clearer. For example, the fact that the tasks were validated by annotators reproducing the reference score in the reasoning gym version was only present in the Appendix but it was a key question about how valid/reviewed are the tasks.
>
> **Response to comment 1:** Thanks for reviewer's suggestion. We have added it in Section 3.
>
> **Comment 2:** It might have been useful to test other scaffolds to ensure that the agents were not substantially worse than they could have otherwise been (e.g. openhands or codex).
>
> **Response to comment 2:** As we mentioned in Appendix F, some parts of our task execution framework refer to the design of OpenHands [1]. The api like `goto_line_action` is just `goto_line` in OpenHands. And the agent will only receive `goto_line` as the action name in its context prompt.
>
> The difference between our tools and OpenHands mostly comes from the command action, since **our ResearchGym supports asynchronous command execution and multi-machine control**. We also changed the file action a little bit to avoid returning too much information and exceeding the context length limit in just one turn.
>
> What's more, when considering the context management part, we have 2 significant improvements:
>
> 1. We improve the system prompt (both the ReAct agent and the corresponding context-summary agent) to make it adjust to the new environment settings.
>
> 2. We change the summary logic to compress the ReAct trajectory based on the context length instead of the number of turns.
>
> **As a result, you can consider our framework as a more advanced version of OpenHands with more powerful tools and reliable context management for research tasks.**
>
> **For codex, it does not support the asynchronous multi-machine control**, and in our settings, the agent needs to deal with at least 2 machines in training (one GPU machine and one CPU machine that can connect to the internet). Therefore, the codex is not suitable for our tasks.
>
> **Comment 3:** One concern is that all performance on this benchmark will be driven by memorization, since all the tasks are based on existing papers that are in training data. It would be more convincing if the authors presented evidence that this was not a concern (i.e. by looking at successes).
>
> **Response to comment 3:** We checked the log and did not find such memorization patterns. And one evidence is that the agents with the ground truth method gain even lower than the baseline (Section 5.3). It proves the agents didn't have the ability to reproduce the original paper. Also, if the agents only memorize the existing repos, they can not gain the full scores (which is 100), as we mentioned in Section 3-Evaluations.
>
>
>
> **Comment 4:** are agents allowed to use the snapshot functionality via tool calls?
>
> **Response to comment 4:**  No, the snapshot time is predefined in our settings. In our experiments, we only save the last step's snapshot.
>
> **Comment 5:** will you release the tasks publicly?
>
> **Response to comment 5:** Yes, we will release the tasks publicly. We have added it to the supplementary material. (We did not provide all datasets and model checkpoints because they are too large, and giving a HuggingFace link will avoid double-blind.)
>
> **Comment 6:**  Typos
>
> **Response to comment 6:** We have corrected the typos in the reference. Thanks for the reviewer's detailed comment.
>
>
> [1] Wang, X., Li, B., Song, Y., Xu, F. F., Tang, X., Zhuge, M., ... & Neubig, G. (2024). Openhands: An open platform for ai software developers as generalist agents. arXiv preprint arXiv:2407.16741.

---

### Official Review · Reviewer_RUwH · 2025-11-01

**Soundness:** 3
**Presentation:** 2
**Contribution:** 3
**Rating:** 6
**Confidence:** 4

**Summary:**

This paper introduces a benchmark of 20 LLM engineering tasks that span the lifecycle of LLM development. Also, a task execution environment is introduced as the framework for running these tasks. In addition, the performance of four models is assessed on these tasks using the ReAct agent, and it is shown that Claude 4 Sonnet achieves the best overall performance and in many of the subtasks, largely from its superior tool use. GPT-5 achieves the best performance in implementing agent scaffoldings, this seems largely due to designing agent scaffoldings that are more resilient to models' failure to use tools.

**Strengths:**

Originality:
- This paper follows the approach of many others in scraping high-quality conference papers for tasks to re-implement.
- The implementation of several agent tools, such as asynchronous command execution, is a simple yet powerful improvement that overcomes one of the most common failure modes I've seen in agent scaffoldings.

Quality:
- Perhaps my biggest source of uncertainty in this paper is the decision to implement an entirely new task execution framework from scratch. This is not an easy problem, and I would feel much more reassured if it had built upon existing battle-tested frameworks.
- There are no confidence intervals on any of the metrics presented, which makes me think that perhaps these are all based on single rollouts, which would really undermine my confidence in the results.
- This paper claims to be unique in that it tests models' creativity, but it follows a similar approach to many other papers in scraping existing conference papers for tasks to implement. It also provides starter code based on those repos. So I'm not sure what is fundamentally unique about this task that allows it to test creativity in ways that others don't.
- Developing 20 new research engineering tasks is a commendable accomplishment. I do have some concern about overlap with tasks in existing benchmarks, given that it also scrapes from conference papers.
- The comparison table also lists multi-GPU, multi-node, and save-and-restore as important differentiators of InnovatorBench, and yet no ablations are performed to show that these features substantially make a difference to model performance. So that's the understanding of the data.
- Similarly, I would appreciate an ablation that shows how important it is to give the agent asynchronous command execution ability, and whether removing that substantially decreased model performance. Because if that doesn't make that big of a difference, it really undermines the motivation for this new, separate framework (and, in any case, it probably could have easily been added to, e.g., https://inspect.aisi.org.uk/)
- Don't get me wrong, it's entirely possible that this was very high-quality evals engineering. It just seems to me to be an instance of undifferentiated heavy lifting.

Clarity:
- The writing is clear and concise, if a bit grandiose.
- In the comparison of different AI benchmarks, it’s shown that InnovatorBench has a “max eval times” of four. I don’t know what “max eval times” means, and it’s not mentioned anywhere else in the paper.

Significance:
-  My guess is that this represents at best a small improvement in the difficulty of research engineering tasks, as demonstrated by the test-time scaling experiment. Though the decision to implement an entirely new framework undermines the community's ability to build on it.

**Weaknesses:**

See strengths section, listed together

**Questions:**

1. Using a simple React agent loop, how did you prevent the agent from submitting early, which is a common failure mode with LLMs?
1. What does it mean to have eval times of 4?
1. You list the time horizon as 2 to 36 hours. What do you mean by that exactly? Is that the amount of time it takes a human to complete the tasks? And do you have any human baseline data?
1. You list multi-node as a key feature of the framework, but it looks like only 1 task makes use of more than one 8× H100 machine. Do you stand by this as an important feature? Or could you remove that task without significant decrease in quality?
1. How many agent runs were performed for each task and model? Do you have any summary statistics? Are the differences between the models statistically significant?

---

> ### Author Response · Authors · 2025-11-17
>
> Thanks for the reviewer's comments. Here are our responses to the comments.
>
> **Comment 1:** Perhaps my biggest source of uncertainty in this paper is the decision to implement an entirely new task execution framework from scratch. This is not an easy problem, and I would feel much more reassured if it had built upon existing battle-tested frameworks.
>
> **Response to comment 1:**
>
> As we mentioned in Appendix F, our task execution framework refers to the design of OpenHands [1]. The api like `goto_line_action` is just `goto_line` in OpenHands. And the agent will only receive `goto_line` as the action name in its context prompt.
>
> The difference between our tools and OpenHands mostly comes from the command action, since ResearchGym supports asynchronous command execution and multi-machine control. We also changed the file action a little bit to avoid returning too much information and exceeding the context length limit in just one turn.
>
>
> What's more, when considering the context management part, we have 2 changes:
>
> 1. We improve the system prompt (both the ReAct agent and the corresponding context-summary agent) to make it adjust to the new environment settings.
>
> 2. We change the summary logic to compress the ReAct trajectory based on the context length instead of the number of turns.
>
> **As a result, you can consider our framework as a more advanced version of OpenHands with more powerful tools and reliable context management for research tasks.**
>
> **Comment 2:** There are no confidence intervals on any of the metrics presented, which makes me think that perhaps these are all based on single rollouts, which would really undermine my confidence in the results.
>
> **Response to comment 2:**
> Thanks for the reviewer's suggestion. We have just added the standard deviation (STD) result of Claude-Sonnet-4 to the result.
>
> | Rollout1 (Final Score) | Rollout1 (Best Score) | Rollout2 (Final Score) | Rollout2 (Best Score) | Rollout3 (Final Score) | Rollout3 (Best Score) | Avg (Final Score) | STD (Final Score) | Avg (Best Score) | STD (Best Score) |
> | ---------------------- | --------------------- | ---------------------- | --------------------- | ---------------------- | --------------------- | ----------------- | ----------------- | ---------------- | ---------------- |
> | 24.01                  | 24.54                 | 23.13                  | 23.15                 | 23.57                  | 23.69                 | 23.57             | 0.440             | 23.79            | 0.70             |
>
> As shown in the table, the STD of the final score is lower than 1, which is lower than the average of the scores. This result reflects that the results are not based on random rollouts.
>
>
>
> **Comment 3:** This paper claims to be unique in that it tests models' creativity, but it follows a similar approach to many other papers in scraping existing conference papers for tasks to implement. It also provides starter code based on those repos. So I'm not sure what is fundamentally unique about this task that allows it to test creativity in ways that others don't.
>
> **Response to comment 3:**  As we mentioned in Table 1, most papers like PaperBench [2] just want the agents to reproduce the result of the paper, instead of asking them to generate new ideas and improve the result.
>
> As a result, a classic but simple agent trajectory should be:
>
> ```
>     Input: A research question or a research topic Q, and an insight memory M=∅
>
>     While the agent is not terminated: //what InnovatorBench focuses on
>         ideas = generate_idea(Q, M)
>         selected_idea = select_idea(ideas)
>         While the selected idea is not finished executing: //debug iteratively, what PaperBench focuses on
>             Generate the code based on the idea
>             Execute the code based on the generated code
>             Verify the code based on the executed result and the idea
>             Decide whether to terminate the executing and debugging process
>
>         // PaperBench finished here, InnovatorBench continues the loop
>
>         Update M with the selected idea
>         Decide whether to terminate the agent
>     Output: The result and the insight gained from the whole research process
> ```
>
> **And the parts outside the PaperBench are the key part of creativity.**
>
> **Comment 4:** Developing 20 new research engineering tasks is a commendable accomplishment. I do have some concern about overlap with tasks in existing benchmarks, given that it also scrapes from conference papers.
>
> **Response to comment 4:**
> Most of the tasks in existing benchmarks are just design tasks that use at most one GPU. However, **most of our work needs about 8 GPUs to run**. So we think there is only a little bit of overlap between our tasks and DatasetResearch [3]. But we asked the agents to train the dataset it finds, and the agent can refine its performance iteratively. These may be the main differences between our tasks and DatasetResearch.

---

> ### Author Response · Authors · 2025-11-17
>
> **Comment 5:** The comparison table also lists multi-GPU, multi-node, and save-and-restore as important differentiators of InnovatorBench, and yet no ablations are performed to show that these features substantially make a difference to model performance. So that's the understanding of the data.
>
> **Response to comment 5:** **Our experimental setup requires substantial computational resources, which makes multi-GPU and multi-node support essential.** Without these capabilities, the experiments simply cannot be executed, and thus no meaningful results would be obtained—effectively yielding a score of zero. **Regarding the save-and-restore feature, it is primarily designed to allow experiments to resume from a checkpoint.** This feature is just for continuing the experiment from the last checkpoint, and **it will not affect the results of the experiment**.
>
> **Comment 6:** Similarly, I would appreciate an ablation that shows how important it is to give the agent asynchronous command execution ability, and whether removing that substantially decreased model performance. Because if that doesn't make that big of a difference, it really undermines the motivation for this new, separate framework (and, in any case, it probably could have easily been added to, e.g., https://inspect.aisi.org.uk/)
>
> **Response to comment 6:** If the asynchronous command execution ability is not supported, the agent will just wait for the command to finish no matter the command is correct or not. So it may cost a lot of time to finish the nonsense command. It will make most of the tasks fail because the agent needs to debug during the execution.
> Besides, if the agent uses a `nohup` to run the command, it needs to use `kill` if the command is not correct. But we found the agent would like to kill the script to support the multi-machine control because it seems like an "unrelated" script. For this reason, we did not grant the agent direct permission to use the `kill` command.
> Thanks for the reviewer's suggestion. We have added a detailed explanation in Appendix F.
>
>
> **Comment 7:** Don't get me wrong, it's entirely possible that this was very high-quality evals engineering. It just seems to me to be an instance of undifferentiated heavy lifting.
>
> **Response to comment 7:** We believe that evaluating the agents in a more complex and realistic environment is very important for the development of the agents.  As the comments mention above, we have added the **asynchronous command execution** and **multi-machine control** to the ResearchGym, which already shows a huge improvement over the existing benchmarks.
>
> **Comment 8:**  Using a simple React agent loop, how did you prevent the agent from submitting early, which is a common failure mode with LLMs?
>
> **Response to comment 8:** Actually, we do not prevent the agent from submitting early.  The agent can submit its output at any time, and it can use the `eval` action to evaluate its output and get the score. In fact, during our experiments, not a single agent submitted early without producing an output that could be evaluated. But we found the agent would like to use `eval` to get the score instead of using the `finish` action to terminate the loop at first.
>
>
> **Comment 9:** What does it mean to have eval times of 4?
>
> **Response to comment 9:** "Max eval times" refers to the maximum number of times an agent can evaluate a given task. The agent can use the `eval` action to assess its output and obtain the score, which can be used to improve and refine its ideas and continue the iterative process (the while loop) above in Comment 3. Once the `eval` action has been called 4 times, the subsequent `eval` action will automatically be replaced by the `finish` action.
> We thank the reviewer for this suggestion and have incorporated the corresponding explanation in Section 3.

---

> ### Author Response · Authors · 2025-11-17
>
> **Comment 10:** You list the time horizon as 2 to 36 hours. What do you mean by that exactly? Is that the amount of time it takes a human to complete the tasks? And do you have any human baseline data?
>
> **Response to comment 10:** No, this is the result generated by Claude rather than by a human. The score represents the average evaluation up to that point in time. For example, if the agent uses the `eval` action to ask for ResearchGym to assess its results at hours 1, 5, 10, and 13 in Task 1, and at hours 2, 10, 23, and 24 in Task 2, then the score at the 16-hour mark would be the average of the score from hour 13 in Task 1 and the score from hour 10 in Task 2.
>
> **Comment 11:** You list multi-node as a key feature of the framework, but it looks like only 1 task makes use of more than one 8× H100 machine. Do you stand by this as an important feature? Or could you remove that task without significant decrease in quality?
>
> **Response to comment 11:** Thanks for reviewer's suggestion.  We believe the ResearchGym should be able to expand to other tasks in the future. Therefore, it is important to realize such feature. It's not just the feature of InnovatorBench. We have added this explanation in Section 4.
>
> **Comment 12:** How many agent runs were performed for each task and model? Do you have any summary statistics? Are the differences between the models statistically significant?
>
> **Response to comment 12:**  Thank you for your comment. The additional experimental results have been provided in our Response to Comment 2.
>
> **We sincerely appreciate the reviewer's constructive suggestions and believe that the additional explanations significantly improve the quality of our submission. We hope that this provides sufficient reasons to raise the score.**
>
> [1] Wang, X., Li, B., Song, Y., Xu, F. F., Tang, X., Zhuge, M., ... & Neubig, G. (2024). Openhands: An open platform for ai software developers as generalist agents. arXiv preprint arXiv:2407.16741.
>
> [2] Starace, G., Jaffe, O., Sherburn, D., Aung, J., Chan, J. S., Maksin, L., ... & Patwardhan, T. (2025). PaperBench: Evaluating AI's Ability to Replicate AI Research. arXiv preprint arXiv:2504.01848.
>
> [3] Li, K., Jiang, M., Fu, D., Wu, Y., Hu, X., Wang, D., & Liu, P. (2025). Datasetresearch: Benchmarking agent systems for demand-driven dataset discovery. arXiv preprint arXiv:2508.06960.

---

### Author Response · Authors · 2025-12-03

We thank the reviewers for their engagement and appreciate their positive feedback: our proposed methods were described as a "powerful improvement", "clear and concise" [RUwH], "realistic" [Hyqf].

The reviewers also raised thoughtful feedback and concerns. In response, we have carefully addressed each point by providing additional experimental results and explaining our design.

Below, we summarize the main concerns from the reviewers and outline how we have addressed them.

- We add the benchmarks in our supplementary material, task_descriptions folder. (Reviewer RUwH, Hyqf, and kpnh)

- We add an explanation in the paper to explain why we do not choose other scaffolds (i.e., Openhands, codex). (Because they do not support asynchronous command execution and multi-machine control, they can not be used in our tasks.) (Reviewer RUwH and Hyqf )

- We have run this benchmark under the same conditions (Claude) 3 times to prove the reproducibility. (Reviewer RUwH)

- Add some details about InnovatorBench and ResearchGym (Reviewer RUwH, Hyqf, and kpnh)

- Add the comparison between InnovatorBench and other benchmarks like PaperBench (Reviewer RUwH and kpnh)

Reviewer RUwH and Hyqf maintain their rating **(6 and 8)** after our rebuttal, and Reviewer kpnh raised its score from 2 to **6** after our rebuttal.

As a result, **all the reviewers gave a positive attitude to our paper.**

---

### Meta-Review · Area_Chair_Fwvn · 2026-01-04

**Summary:**

In all, the reviewers have converged to a positive attitude towards the paper, and thus I recommend acceptance.

Below is a brief summary of the main concerns raised by the reviewers:

Reviewer RUwH:
1. Lacking confidence interval: the authors have added it during their rebuttal.
2. Uniqueness in testing models' creativity: the authors provide pseudocode to explain.
3 Implementation details: clarified later by the authors

Reviewer Hyqf:
1. Testing other scaffolds: The authors added an explanation in the paper to explain why we do not choose other scaffolds

Reviewer kpnh:
1. Limited number of tasks included in InnovatorBench: the authors explained that annotation process is costly.
2. Lack of empirical results of open-sourced light-weight LLMs, such as Qwen3-32B. The authors have added the Qwen3-32b experiment.

In all, I think the authors have provided a good rebuttal and addressed most of the concerns.

**Reviewer Concerns:**

RUwH: I think the authors have addressed/cleared most concerns raised by the reviewer, but I also feel that the uniqueness in testing models' creativity is limited.
Reviewer Hyqf: the authors have addressed all the concerns.
Reviewer kpnh: the authors have addressed all the concerns.

**Reviewer Scores:**

RUwH and Hyqf will likely maintain their scores (6 and 8). The concerns raised by kpnh were relatively minor, have been addressed by the authors, and the reviewer has agreed to raise the score to 6.

---

### Decision · Program_Chairs · 2026-01-26

Accept (Poster)